# KAKURENBO: Adaptively Hiding Samples in Deep Neural Network Training

Truong Thao Nguyen[1]    Balazs Gerofi[2,4]    Edgar Josafat Martinez-Noriega[1]
François Trahay[3]    Mohamed Wahib[4]

[1] National Institute of Advanced Industrial Science and Technology (AIST), Japan
[2] Intel Corporation, USA
[3] Télécom SudParis, Institut Polytechnique de Paris, France
[4] RIKEN Center for Computational Science, Japan

```
{nguyen.truong, edgar.martineznoriega}@aist.go.jp
             balazs.gerofi@intel.com
      francois.trahay@telecom-sudparis.eu
             mohamed.attia@riken.jp
```

## Abstract

This paper proposes a method for hiding the least-important samples during the training of deep neural networks to increase efficiency, i.e., to reduce the cost of training. Using information about the loss and prediction confidence during training, we adaptively find samples to exclude in a given epoch based on their contribution to the overall learning process, without significantly degrading accuracy. We explore the converge properties when accounting for the reduction in the number of SGD updates. Empirical results on various large-scale datasets and models used directly in image classification and segmentation show that while the with-replacement importance sampling algorithm performs poorly on large datasets, our method can reduce total training time by up to 22% impacting accuracy only by 0.4% compared to the baseline. Code available at https://github.com/TruongThaoNguyen/kakurenbo

## 1   Introduction

Empirical evidence shows the performance benefits of using larger datasets when training deep neural networks (DNN) for computer vision, as well as in other domains such as language models or graphs [1]. More so, attention-based models are increasingly employed as pre-trained models using unprecedented dataset sizes, e.g. the JFT-3B dataset consists of nearly three billion images, annotated with a class-hierarchy of around 30K labels [2], LIAON-5B provides 5,85 billion CLIP-filtered image-text pairs that constitute over 240TB [3]. A similar trend is also observed in scientific computing, e.g., DeepCAM, a climate simulation dataset, is over 8.8TB in size [4]. Furthermore, the trend of larger datasets prompted efforts that create synthetic datasets using GANS [5] or fractals [6]. The downside of using large datasets is, however, the ballooning cost of training. For example, it has been reported that training models such as T5 and AlphaGo cost $1.3M [7] and $35M [8], respectively. Additionally, large datasets can also stress non-compute parts of supercomputers and clusters used for DNN training (e.g., stressing the storage system due to excessive I/O requirements [9, 10]).

In this paper, we are focusing on accelerating DNN training over large datasets and models. We build our hypothesis on the following observations on the effect of sample quality on training: a) *biased with-replacement sampling* postulates that not all samples are of the same importance and a biased, with-replacement sampling method can lead to faster convergence [11, 12], b) *data pruning* methods show that when select samples are pruned away from a dataset, the predication accuracy

Table 1: Summary of related works. Complexity on the number of samples $N$, number of Epochs $M$. and ensemble size $E$.

| Approach | Method | Merits (+)
Demerits (-) | Online/
Offline | Practical Overhead
(Bottleneck) | Complexity
(Cost) |
|---|---|---|---|---|---|
| **Biased w/
Replacement
Sampling** | Importance
Sampling [11] | + Theoretically faster convergence
- No demonstrated speedup on large datasets (Section 4)
- Nondeterministic | Online | Sorting samples | $O(N.log(N))$ |
| | RHO-LOSS [12] | + Theoretically faster convergence
- No demonstrated speedup
- Nondeterministic | | Hold-out approx. | $O(N^2)$ |
| **Data
Pruning**
(prune dataset
offline to save cost in
future training) | Forgiveness
Scores [13] | + Robust
- Full training needed to identify samples to prune | Offline | Tracking change in
prediction | $O(N \cdot M^2)$ |
| | EL2N [15] | + Demonstrated speedup
- Full training needed to identify samples to prune
- Limited scalability | | Sorting samples | $O(N^2)$ |
| | Memorization [14] | + Demonstrated speedup
- Full training needed to identify samples to prune
- Limited scalability | | Tracking cross-sample
prediction | $O(N^2)$ |
| | Ensemble Active
Learning [16] | + Demonstrated speedup
- Full training needed to identify samples to prune
- Limited scalability | | Model uncertainty
approximation | $O(N^2)$ |
| | Diverse Ensembles
(DDD) [20] | + Demonstrated speedup
- Full training needed to identify samples to prune
- Limited scalability | | Tracking cross-ensemble
prediction | $O(N^2 \cdot E)$ |
| **Hiding Samples** | Selective
Backprop [17] | - Arbitrary hiding samples
- No convergence guarantee
- No demonstrated speedup while maintaining accuracy | Online | Sorting samples | $O(N \cdot log(N))$ |
| | GRAD-MATCH
[21] | + Adpative
- Limited to single GPU (distributed training impractical)
- No convergence guarantee for skipping selection | | Matching samples to
gradients | $O(N.M)$ |
| | **This Work** | + Scalable
+ Efficiently hiding samples
+ Theoretically convergence guarantee
+ Demonstrated speedup while maintaining accuracy | | Sorting samples | $O(N \cdot log(N))$ |

that can be achieved by training from scratch using the pruned dataset is similar to that of the original dataset [13, 14, 15, 16]. Our hypothesis is that if samples have a varying impact on the learning process and their impact decreases as the training progresses, then we can in real-time, adaptively, exclude samples with the least impact from the dataset during neural network training.

In this paper, we dynamically hide samples in a dataset to reduce the total amount of computing and the training time, while maintaining the accuracy level. Our proposal, named KAKURENBO, is built upon two pillars. First, using combined information about the loss and online estimation of the historical prediction confidence (see Section 3.1) of input samples, we adaptively exclude samples that contribute the least to the overall learning process on a per-epoch basis. Second, in compensation for the decrease in the number of SGD steps, we derive a method to dynamically adjust the learning rate and the upper limit on the number of samples to hide in order to recover convergence rate and accuracy.

We evaluate performance both in terms of reduction in wall-clock time and degradation in accuracy. Our main results are twofold: first, we show that decaying datasets by eliminating the samples with the least contribution to learning has no notable negative impact on the accuracy and convergence and that the overhead of identifying and eliminating the least important samples is negligible. Second, we show that decaying the dataset can significantly reduce the total amount of computation needed for DNN training. We also find that state-of-the-art methods such as importance sampling algorithm [11], pruning [13], or sample hiding techniques [17, 18] performs poorly on large-scale datasets. To the contrary, our method can reduce training time by $10.4\%$ and $22.4\%$ on ImageNet-1K [19] and DeepCAM [4], respectively, impacting Top-1 accuracy only by $0.4\%$.

## 2 Background and Related Work

As the size of training datasets and the complexity of deep-learning models increase, the cost of training neural networks becomes prohibitive. Several approaches have been proposed to reduce this training cost without degrading accuracy significantly. Table 1 summarizes related work against this proposal. This section presents the main state-of-the-art techniques. Related works are detailed in the Appendix-E.

**Biased with-Replacement Sampling** has been proposed as a method to improve the convergence rate in SGD training [11, 12]. Importance sampling is based on the observation that not all samples are of equal *importance* for training, and accordingly replaces the regular uniform sampling used to draw samples from datasets with a biased sampling function that assigns a likelihood to a sample being

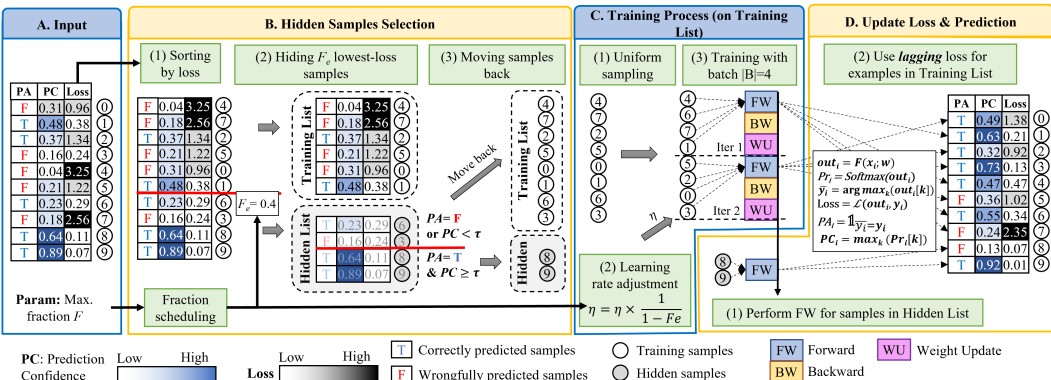

Figure 1: **Overview of KAKURENBO**. At each epoch, samples are filtered into two different subsets, the training list and the hidden list, based on their loss, prediction accuracy (PA), and prediction confidence (PC), with a maximum hidden fraction of $F$. PA and PC are used to drive sample move back decisions. Samples in the training list are processed using uniform sampling without replacement. The loss and the prediction accuracy, calculated from the training process, are reused to filter samples in the next epoch. For samples on the hidden list, KAKURENBO only calculates the loss and PA by performing the forward pass at the end of each epoch.

drawn proportional to its importance; the more important the sample is, the higher the likelihood it would be selected. The with-replacement strategy of importance sampling maintains the total number of samples the network trains on. Several improvements over importance sampling have been proposed for distributed training [22], or for estimating the importance of samples [12, 23, 24, 25, 26].

Overall, biased with-replacement sampling aims at increasing the convergence speed of SGD by focusing on samples that induce a measurable change in the model parameters, which would allow a reduction in the number of epochs. While these techniques promise to converge in fewer epochs on the whole dataset, each epoch requires computing the importance of samples which is time-consuming.

**Data Pruning techniques** are used to reduce the size of the dataset by removing less important samples. Pruning the dataset requires training on the full dataset and adds significant overheads for quantifying individual differences between data points [27]. However, the assumption is that the advantage would be a reduced dataset that replaces the original datasets when used by others to train. Several studies investigate the selection of the samples to discard from a dataset[13, 15, 14, 16] [28].

Pruning the dataset does reduce the training time without significantly degrading the accuracy [13, 14]. However, these techniques require fully training the model on the whole dataset to identify the samples to be removed, which is compute intensive.

**Selective-Backprop** [17] combines importance sampling and online data pruning. It reduces the number of samples to train on by using the output of each sample's forward pass to estimate the sample's importance and cuts a fixed fraction of the dataset at each epoch. While this method shows notable speedups, it has been evaluated only on tiny datasets without providing any measurements on how accuracy is impacted. In addition, the authors allow up to 10% reduction in test error in their experiments.

**Grad-Match** [18] is an online method that selects a subset of the samples that would minimize the gradient matching error. The authors approximate the gradients by only using the gradients of the last layer, use a per-class approximation, and run data selection every $R$ epochs, in which case, the same subsets and weights will be used between epochs. Due to the infrequent selection of samples, Grad-Match often needs a larger number of epochs to converge to the same validation accuracy that can be achieved by the baseline [29]. Moreover, Grad-Match is impractical in distributed training, which is a de facto requirement in large dataset and models. Distributed Grad-Match would require very costly collective communication to collect the class approximations and to do the matching optimization. This is practically a very high cost for communication per epoch that could even exceed the average time per epoch.

## 3 KAKURENBO: Adaptively Hiding Samples

In this work, we reduce the amount of work in training by adaptively choosing samples to hide in each epoch. We consider a model with a loss function $\ell(\mathbf{w}, \mathbf{x}_n, \mathbf{y}_n)$ where $\{\mathbf{x}_n, \mathbf{y}_n\}_{n=1}^{N}$ is a dataset of $N$ sample-label pairs ($x_n \in X$), and $G : X \to X$ is a function that is applied to hide certain samples during training, e.g., by ranking and cut-off some samples. Using SGD with a learning-rate $\eta$ and batch size of $B$, the update rule for each batch when training with original full dataset is

$$\mathbf{w}_{t+1} = \mathbf{w}_t - \eta \frac{1}{B} \sum_{n \in \mathcal{B}(k(t))} \nabla_{\mathbf{w}} \ell(\mathbf{w}_t, \mathbf{x}_n, \mathbf{y}_n) \tag{1}$$

where $k(t)$ is sampled from $[N/B] \triangleq \{1, \ldots, N/B\}$, $\mathcal{B}(k)$ is the set of samples in batch $k$ (to simplify, $B$ is divisible by $N$). We propose to hide $M$ examples by applying the a hiding function $G$. We modify the learning rule to be

$$\mathbf{w}_{t+1} = \mathbf{w}_t - \eta \frac{1}{B} \sum_{n \in \mathcal{B}(k(t))} \nabla_{\mathbf{w}} \ell(\mathbf{w}_t, G(\mathbf{x}_n), \mathbf{y}_n) \tag{2}$$

using $B$ batch at each step, which is composed of $N/B$ steps. Since we exclude $M$ samples, the aggregate number of steps is reduced from $N/B$ to become $(N - M)/B$, i.e., fixing the batch size and reducing the number of samples reduces the number of SGD iterations that are performed for each epoch.

Sample hiding happens before presenting the input to each epoch. The training set that excludes the hidden samples $(N - M)$ is then shuffled for the training to process with the typical w/o replacement uniform sampling method.

Based on the above training strategy, we propose KAKURENBO, a mechanism to dynamically reduce the dataset during model training by selecting important samples. The workflow of our scheme is summarized in Figure 1. First, (B.1) we sort the samples of a dataset according to their loss. We then (B.2) select a subset of the dataset by *hiding* a fixed fraction $F$ of the data: the samples with the lowest loss are removed from the training set. Next, (B.3) hidden samples that maintain a correct prediction with high confidence (see Section 3.1) are moved back to the epoch training set. The training process (C) uses uniform sampling without replacement to pick samples from the training list. KAKURENBO adapts the learning rate (C.2) to maintain the pace of the SGD. At the end of the epoch, we perform the forward pass on samples to compute their loss and the prediction information on the up-to-date model (D). However, because calculating the loss for all samples in the dataset is prohibitively compute intensive [11], we propose to reuse the loss computed during the training process, which we call *lagging* loss (D.2). We only recompute the loss of samples from the hidden list (D.1). In the following, we detail the steps of KAKURENBO.

### 3.1 Hidden Samples Selection

We first present our proposed algorithm to select samples to hide in each epoch. We follow the observation in [11] that not all the samples are equal so that not-too-important samples can be hidden during training. An important sample is defined as the one that highly contributes to the model update, e.g., the gradient norm $\nabla_{\mathbf{w}} \ell(\mathbf{w}_t, \mathbf{x}_n, \mathbf{y}_n)$ in Equation 1. Removing the fraction $F$ of samples with the least impact on the training model from the training list could reduce the training time, i.e., the required computing resource, without affecting the convergence of the training process. Selecting the fraction $F$ is arbitrary and driven by the dataset/model. If the fraction $F$ is too high, the accuracy could drop. In contrast, the performance gained from hiding samples will be limited if $F$ is small, or potentially less than the overhead to compute the importance of samples. In this work, we aim to design an adaptive method to select the fraction $F^*$ in each epoch. We start from a tentative maximum fraction $F$ at the beginning of the training process. We then carefully select the hidden samples from $F$ based on their importance and then move the remaining samples back to the training set. That is, at each epoch a dynamic hiding fraction $F^*$ is applied.

It is worth noting that the maximum fraction number $F$ does not need to be strictly accurate in our design; it is a maximum ceiling and not the exact amount of samples that will be hidden. However, if the negative impact of hiding samples, i.e. $\frac{\sum_{n \in 1}^{F^* \times N} \|\nabla_{\mathbf{w}} \ell(\mathbf{w}_t, \mathbf{x}_n, \mathbf{y}_n)\|}{\sum_{n \in 1}^{N} \|\nabla_{\mathbf{w}} \ell(\mathbf{w}_t, \mathbf{x}_n, \mathbf{y}_n)\|}$, becomes too high, it could

significantly affect the accuracy. For example, when a high maximum fraction $F$ is set and/or when most of the samples have nearly the same absolute contribution to the update, e.g., at the latter epoch of the training process. We investigate how to choose the maximum hiding fraction in each epoch in Section 3.3.

**Moving Samples Back:** since the loss is computed in the forward pass, it is frequently used as the metric for the importance of the sample, i.e. samples with high loss contribute more to the update and are thus important [11, 22]. However, the samples with the smallest loss do not necessarily have the least impact (i.e., gradient norm) on the model, which is particularly true at the beginning of the training, and removing such high-impact samples may hurt accuracy. To mitigate the misselection of important samples as unimportant ones, we propose an additional rule to filter the low-loss samples based on the observation of historical prediction confidence [13]. The authors in [13] observed that some samples have a low frequency of toggling back from being classified correctly to incorrectly over the training process. Such samples can be pruned from the training set eternally. Because estimating the per-sample prediction confidence before training (i.e., offline) is compute-intensive, in this work, we perform an online estimation to decide whether an individual sample has a history of correct prediction with high confidence or not in a given epoch. Only samples that have low loss and sustain correct prediction with high confidence in the current epoch are hidden in the following epoch.

A sample is correctly predicted with high confidence at an epoch $e$ if it is predicted correctly (**PA**) and the prediction confidence (**PC**) is no less than a threshold $\tau$, which we call the *prediction confidence threshold*, at the previous epoch. In addition to the prediction confidence of a given sample $(x, y)$ is the probability that the model predicts this sample to map to label $y$:

$$
\begin{aligned}
out &= model(\mathbf{w}_e, x, y) \\
PC &= \max_k(\sigma(out_k))
\end{aligned}
\tag{3}
$$

where $\sigma$ is a sigmod (softmax) activation function. In this work, unless otherwise mentioned, we set the prediction confidence threshold to $\tau = 0.7$ as investigated in Section 4.3.

## 3.2 Reducing the Number of Iterations in Batch Training: Learning Rate Adjustment

After hiding samples, KAKURENBO uses uniform without replacement sampling to train on the remaining samples from the training set. In this section, we examine issues related to convergence when reducing the number of samples and we provide insight into the desirable convergence properties of adaptively hiding examples.

Implicit bias in the SGD training process may lead to convergence problems [30]: when reducing the total number of iterations at fixed batch sizes, SGD selects minima with worse generalization. We examine the selection mechanism in SGD when reducing the number of iterations at a fixed batch size. For optimizations of the original datasets, i.e., without example hiding, we use loss functions of the form

$$
f(\mathbf{w}) = \frac{1}{N} \sum_{n=1}^{N} \ell(\mathbf{w}, \mathbf{x}_n, \mathbf{y}_n) ,
\tag{4}
$$

where $\{\mathbf{x}_n, \mathbf{y}_n\}_{n=1}^{N}$ is a dataset of $N$ data example-label pairs and $\ell$ is the loss function. We use SGD with batch of size $B$ and learning-rate $\eta$ with the update rule

$$
\mathbf{w}_{t+1} = \mathbf{w}_t - \eta \frac{1}{B} \sum_{n \in \mathcal{B}(k(t))} \nabla_{\mathbf{w}} \ell(\mathbf{w}_t, \mathbf{x}_n, \mathbf{y}_n) .
\tag{5}
$$

for without replacement sampling, $B$ divisible by $N$ (to simplify), and $k(t)$ sampled uniformly from $\{1, \ldots, N/B\}$. When using an over-parameterized model as is the case with deep neural networks, we typically converge to a minimum $\mathbf{w}^*$ that is a global minimum on all data points $N$ in the training set [31, 14]. Following Hoffer et al. [32], linearizing the dynamics of Eq. 5 near $\mathbf{w}^*$ $(\forall n : \nabla_{\mathbf{w}} \ell(\mathbf{w}^*, \mathbf{x}_n, \mathbf{y}_n) = 0)$ gives

$$
\mathbf{w}_{t+1} = \mathbf{w}_t - \eta \frac{1}{B} \sum_{n \in \mathcal{B}(k(t))} \mathbf{H}_n \mathbf{w}_t ,
\tag{6}
$$

where we assume $\mathbf{w}^* = 0$ since the models we target are over-parameterized (i.e., deep networks) leading to converge to a minimum $\mathbf{w}^*$. We also assume $\mathbf{H}_n \triangleq \nabla^2_{\mathbf{w}} \ell(\mathbf{w}, \mathbf{x}_n, \mathbf{y}_n)$ represents the per-example loss Hessian. SGD can select only certain minima from the many potential different global minima for the loss function of a given the full training set $N$ (and without loss of generality, for the training dataset after hiding samples $N - M$). The selection of minima by SGD depends on the batch sizes and learning rate through the averaged Hessian over batch $k$

$$\langle \mathbf{H} \rangle_k \triangleq \frac{1}{B} \sum_{n \in \mathcal{B}(k)} \mathbf{H}_n$$

and the maximum over the maximal eigenvalues of $\{\langle \mathbf{H} \rangle_k\}_{k=1}^{N/B}$

$$\lambda_{\max} = \max_{k \in [N/B]} \max_{\forall \mathbf{v} : \|\mathbf{v}\| = 1} \mathbf{v}^\top \langle \mathbf{H} \rangle_k \mathbf{v}. \tag{7}$$

This $\lambda_{\max}$ affects SGD through the Theorem proved by Hoffer et al. [32]: the iterates of SGD (Eq. 6) will converge if

$$\lambda_{\max} < \frac{2}{\eta}$$

The theorem implies that a high learning rate leads to convergence to be for global minima with low $\lambda_{\max}$ and low variability of $\mathbf{H}_n$. Since in this work we are fixing the batch size, we maintain $\lambda_{\max}$, the variability of $\langle \mathbf{H} \rangle_k$. Therefore, certain minima with high variability in $\mathbf{H}_n$ will remain accessible to SGD. Now SGD may converge to these high variability minima, which were suggested to exhibit worse generalization performance than the original minima [33].

We mitigate this problem by reducing the delta by which the original learning rate decreases the learning rate (after the warm-up phase [34]). That way we make these new minima inaccessible again while keeping the original minima accessible. Specifically, KAKURENBO adjusts the learning rate at each epoch (or each iteration) $e$ by the following rule:

$$\eta_e = \eta_{base,e} \times \frac{1}{1 - F_e} \tag{8}$$

where $\eta_{base,e}$ is the learning rate at epoch $e$ in the non-hiding scenario and $F_e$ is the hiding fraction at epoch $e$. By multiplying the base learning rate with a fraction $\frac{1}{1-F_e}$, KAKURENBO is independent of the learning rate scheduler of the baseline scenario and any other techniques related to the learning rate.

### 3.3 Adjusting the Maximum Hidden Fraction $F$

Merely changing the learning rate may not be sufficient, when some minima with high variability and low variability will eventually have similar $\lambda_{\max}$, so SGD will not be able to discriminate between these minima.

To account for this, we introduce a schedule to reduce the maximum hidden fraction. For the optimum of the set of hidden samples, $\mathbf{w_M} = G(\mathbf{x}_n)$ and an overall loss function $F(\cdot)$ that acts as a surrogate loss for problems which are sums of non-convex losses $f_i(\mathbf{w})$, where each is individually non-convex in $\mathbf{w}$. With Lipschitz continuous gradients with constant $L_i$ we can assume

$$\|\nabla f_i(\mathbf{w_1}) - \nabla f_i(\mathbf{w_2})\| \leq L_i \|\mathbf{w_1} - \mathbf{w_2}\|$$

Since we are hiding samples when computing the overall loss function $F(\cdot)$, we assume each of the functions $f_i(.)$ shares the same minimum value $\min_{\mathbf{w}} f_i(\mathbf{w}) = \min_{\mathbf{w}} f_j(\mathbf{w}) \ \forall \ i, j$. We extend the proof of the theorem on the guarantees for a linear rate of convergence for smooth functions with strong convexity [35] to the non-convex landscape obtained when training with hidden samples (proof in Appendix A)

**Lemma 1.** *Let $F(\mathbf{w}) = \mathbb{E}[f_i(\mathbf{w})]$ be non-convex. Set $\sigma^2 = \mathbb{E}[\|\nabla f_i(\mathbf{w_M})\|^2]$ with $\mathbf{w}^* := argminF(\mathbf{w})$. Suppose $\eta \leq \dfrac{1}{\sup_i L_i}$. Let $\Delta_t = \mathbf{w_t} - \mathbf{w}$. After $T$ iterations, SGD satisfies:*

$$\mathbb{E}\left[\|\Delta_T\|^2\right] \leq (1 - 2\eta\hat{C})^T \|\Delta_0\|^2 + \eta R_\sigma \tag{9}$$

*where $\hat{C} = \lambda(1 - \eta \sup_i L_i)$ and $R_\sigma = \dfrac{\sigma^2}{\hat{C}}$.*

Since the losses $f_i(\mathbf{w})$ are effectively dropping for individual samples, driven by the weight update, we thus drop the maximum fraction that can be hidden to satisfy Eq. 9. Specifically, we suggest selecting a reasonable number that is not too high at the first epoch, e.g, $F = 0.3$. We then adjust the maximum fraction per epoch (denoted as $F_e$) to achieve $F_e$. We suggest using step scheduling, i.e., to reduce the maximum hiding fraction gradually with a factor of $\alpha$ by the number of epochs increases. For example, we set $\alpha$ as [1, 0.8, 0.6, 0.4] at epoch [0, 30, 60, 80] for ImageNet-1K and [0, 60, 120, 180] for CIFAR-100, respectively.

### 3.4 Update Loss and Prediction

Our technique is inspired by an observation that the importance of each sample of the local data does not change abruptly across multiple SGD iterations [22]. We propose to reuse the loss and historical prediction confidence, computed during the training process, and only recompute those metrics for samples from the hidden list. Specifically, the loss and historical prediction confidence of samples are computed only one time at each epoch, i.e., when the samples are fed to the forward pass. It is not re-calculated at the end of each epoch based on the latest model. Therefore, only samples of the last training iteration of a given epoch have an up-to-date loss. Furthermore, if we re-calculate the loss of hidden samples, i.e., only skip the backward and weight update pass of these samples, the loss of hidden samples is also up-to-date. For instance, if we cut off 20% of samples, we have nearly 20% up-to-date losses and 80% of not-up-to-date losses at the end of each epoch As the result, in comparison to the baseline scenario, KAKURENBO helps to reduce the total backward and weight update time by a fraction of $F_e$ while it does not require any extra forward time

## 4 Evaluation

We evaluate KAKURENBO using several models on various datasets. We measure the effectiveness of our proposed method on two large datasets. We use Resnet50 [36] and EfficientNet [37] on ImageNet-1K [19], and DeepCAM [4], a scientific image segmentation model with its accompanying dataset. To confirm the correctness of the baseline algorithms we also use WideResNet-28-10 on the CIFAR-100 dataset. Details of experiment settings and additional experiments such as ablation studies and robustness evaluation are reported in Appendix-B and Appendix-C. We compare the following training strategies:

- **Baseline**: We follow the original training regime and hyper-parameters suggested by their authors using uniform sampling without replacement.

- **Importance Sampling With Replacement** [11] (**ISWR**): In each iteration, each sample is chosen with a probability proportional to its loss. The with-replacement strategy means that a sample may be selected several times during an epoch, and the total number of samples fed to the model is the same as the baseline implementation.

- **FORGET** is an online version of a pruning technique [13]: instead of fully training the model using the whole dataset before pruning, we train it for 20 epochs, and a fraction $F$ of forgettable samples (i.e. samples that are always correctly classified) are pruned from the dataset[1]. The training then restarts from epoch 0. We report the total training time that includes the 20 epochs of training with the whole dataset, and the full training with the pruned dataset.

- **Selective Backprop (SB)** [17] prioritizes samples with high loss at each iteration. It performs the forward pass on the whole dataset, but only performs backpropagation on a subset of the dataset.

- **Grad-Match** [18] trains using a subset of the dataset. Every $R$ epoch, a new subset is selected so that it would minimize the gradient matching error.

- **KAKURENBO**: our proposed method where samples are hidden dynamically during training.

It is worth noting that we follow the hyper-parameters reported in [38] for training ResNet-50, [39] for training WideResNet-28-10, [37] for training EfficientNet-b3, and [4] for DeepCAM. We show the detail of our hyper-parameters in Appendix B. We configure ISWR, and FORGET to remove the same fraction $F$ as KAKURENBO. For SB, we use the $\beta = 1$ parameter that results in removing $50\%$ of samples. Unless otherwise mentioned, our default setting for the maximum hidden fraction

---

[1]We choose the samples to remove by increasing number of forgetting events as in [13].

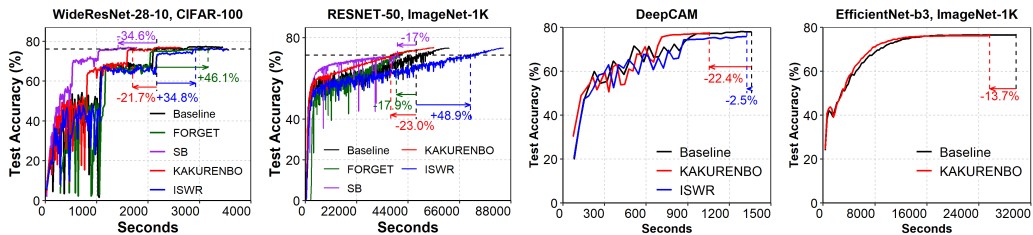

Figure 2: Convergence and speedup of KAKURENBO and importance sampling (ISWR).

Table 2: Max testing accuracy (Top-1) in percentage of KAKURENBO in the comparison with those of the Baseline and other SOTA methods. **Diff.** represent the gap to the Baseline.

| Setting | CIFAR-100 WRN-28-10 | | ImageNet-1K ResNet-50 | | EfficientNet-b3 | | DeepCAM | |
|---|---|---|---|---|---|---|---|---|
| | Acc. | Diff. | Acc. | Diff. | Acc. | Diff. | Acc. | Diff. |
| Baseline | 77.49 | | 74.89 | | 76.63 | | 78.14 | |
| ISWR | 76.51 | (-0.98) | 74.91 | (+0.02) | N/A | | 75.75 | (-2.39) |
| FORGET | 76.14 | (-1.35) | 73.70 | (-1.20) | N/A | | N/A | |
| SB | 77.03 | (-0.46) | 71.37 | (-3.52) | N/A | | N/A | |
| KAKURENBO | 77.21 | (-0.28) | 75.15 | (+0.26) | 76.23 | (-0.5) | 77.42 | (-0.9) |

Table 3: Comparison with Grad-Match in a single GPU (cutting fraction is set to 0.3.

| Setting | CIFAR-100 ResNet-18 | |
|---|---|---|
| | Acc. | Time (sec) |
| Baseline | 77.98 | 8556 |
| Grad-Match-0.3 | 76.87 (-1.11) | 8104 (-5.3%) |
| KAKURENBO-0.3 | 77.05 (-0.93) | 8784 (+2.7%) |

$F$ for KAKURENBO is 30%, except for the CIFAR-100 small dataset, for which we use 10% (see below).

To maintain fairness in comparisons between KAKURENBO and other state-of-the-art methods, we use the same model and dataset with the same hyper-parameters. This would mean we are not capable of using state-of-the-art hyper-parameters tuning methods to improve the accuracy of ResNet-50/ImageNet (e.g., as in [40]). That is since the state-of-the-art hyper-parameters tuning methods are not applicable to some of the methods we compare with. Particularly, we can not apply GradMatch for training with a large batch size on multiple GPUs. Thus, we compare KAKURENBO with GradMatch using the setting reported in [18], i.e., CIFAR-100 dataset, ResNet-18 model.

## 4.1 Accuracy

The progress in the top-1 test accuracy with a maximum hiding fraction of 0.3 is shown in Figure 2. Table 2 summarizes the final accuracy for each experiment. We present data on the small dataset of CIFAR-100 to confirm the correctness of our implementation of ISWR, FORGET, and SB. Table 3 reports the single GPU accuracy obtained with Grad-Match because it cannot work on distributed systems. For CIFAR-100, we report similar behavior as reported in the original work on ISWR [11], SB [17], FORGET [13], and Grad-Match [18]: ISWR, FORGET, and Grad-Match degrade accuracy by approximately 1%, while SB and KAKURENBO roughly perform as the baseline. KAKURENBO on CIFAR-100 only maintains the baseline accuracy for small fractions (e.g. $F = 0.1$). When hiding a larger part of the dataset, the remaining training set becomes too scarce, and the model does not generalize well.

On the contrary, on large datasets such as ImageNet-1K, ISWR and KAKURENBO slightly improve accuracy (by 0.2) in comparison to the baseline, while FORGET and SB degrade accuracy by 1.2% and 3.5%, respectively. On DeepCAM, KAKURENBO does not affect the accuracy while ISWR degrades it by 2.4% in comparison to the baseline[2]. Table 4 reports the accuracy obtained for transfer learning. We do not report Grad-Match results because we could not apply it to this application. Using SB significantly degrades accuracy compared to the baseline, while ISWR, FORGET, and KAKURENBO maintains the same accuracy as the baseline. Especially, as reported in Figure 3, the testing accuracy obtained by KAKURENBO are varied when changing the maximum hiding fraction. We observe that for small hiding fractions, KAKURENBO achieves the same accuracy as

---

[2]We confirm the same behaviors of KAKURENBO and other methods with different hyper-parameters (as shown in Appendix-C).

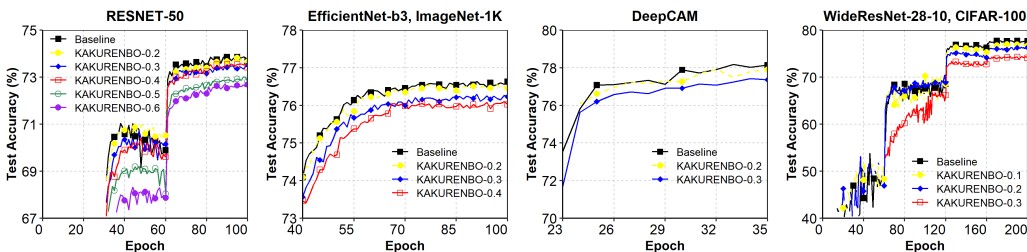

Figure 3: Test accuracy vs. epoch of KAKURENBO with different maximum hiding fractions $F$.

Table 4: Impact of KAKURENBO in transfer learning with DeiT-Tiny-224 model.

|  | Dataset | Metrics | Baseline | ISWR | FORGET | SB | KAKUR. |
|---|---|---|---|---|---|---|---|
| Up stream | Fractal-3K | Loss | 3.26 | 3.671 | 3.27 | 4.18 | 3.59 |
|  |  | Time (min) Impr. | 623 - | 719 (+15.4%) | 533 (-14.4%) | 414 (-33.5%) | 529 (-15.1%) |
| Down stream | CIFAR-10 | Acc. (%) Diff. | 95.03 - | 95.79 (+0.76) | 95.85 (+0.82) | 93.59 (-1.44) | 95.28 (+0.25) |
|  | CIFAR-100 | Acc. (%) Diff. | 79.69 - | 79.62 (-0.07) | 79.95 (+0.26) | 76.98 (-2.71) | 79.35 (-0.34) |

Table 5: Impact of $\tau$ (prediction confidence threshold) on the performance of KAKURENBO.

| Setting | CIFAR-100 WRN-28-10 | |
|---|---|---|
|  | Acc. | Time (sec) |
| $\tau = 0.5$ | 76.37 | 753.9 |
| $\tau = 0.7$ | 76.81 | 758.9 |
| $\tau = 0.9$ | 76.92 | 760.7 |

the baseline. When increasing hiding fractions, as expected, the degradation of the testing accuracy becomes more significant.

## 4.2 Convergence Speedup and Training Time

Here we discuss KAKURENBO's impact on training time. Figure 2 reports test accuracy as the function of elapsed time (note the X-axis), and reports the training time to a target accuracy. Table 4 reports the upstream training time of DeiT-Tiny-224. The key observation of these experiments is that KAKURENBO reduces the training time of Wide-ResNet by 21.7%, of ResNet-50 by 23%, of EfficientNet by 13.7%, of DeepCAM by 22.4%, and of DeiT-Tiny by 15.1% in comparison to the baseline training regime.

Surprisingly, Importance Sampling With Replacement (ISWR) [11] introduces an overhead of 34.8% on WideNet, of 41% on ImageNet-1K and offers only a slight improvement of 2.5% on DeepCAM. At each epoch, ISWR processes the same number of samples as the baseline. Yet, it imposes an additional overhead of keeping track of the importance (i.e., the loss) of all input samples. While on DeepCAM it achieves a modest speedup due to its faster convergence, these experiments reveal that ISWR's behavior is widely different on large datasets than on the smaller ones previously reported [11, 17].

FORGET increases the training time of WideResNet by 46.1% because of the additional 20 epochs training on the whole dataset needed for pruning the samples. When the number of epoch is large, such as for ResNet50 that runs for 600 epochs, FORGET decreases the training time by 17.9%, and for DeiT by 14.4%. However, this reduction of training time comes at the cost of degradation of the test accuracy. On WideResNet and ResNet, SB performs similarly to KAKURENBO by reducing the training time without altering the accuracy. However, SB significantly degrades accuracy compared to the baseline for ImageNet and DeiT.

It is worth noting that KAKURENBO has computation overheads for updating the loss and prediction (Step D in Figure 1), and sorting the samples based on the loss (Step A in Figure 1). For example, Figure 4 reports the measured speedup per epoch as compared to the baseline epoch duration. The speedup follows the same trend as the hiding rate. This is because reducing the number of samples in the training set impacts the speed of the training. The measured speedup does not reach the maximum hiding rate because of the computation overhead. The performance gain from hiding samples will be limited if the maximum hiding fraction $F$ is small, or potentially less than the overhead to compute the importance score of samples. In experiments using multiple GPUs, those operations are performed in parallel to reduce the running time overhead. When using a single GPU on CIFAR-100 with ResNet-18 (Table 3), the computational overhead is bigger than the speedup gained from hiding

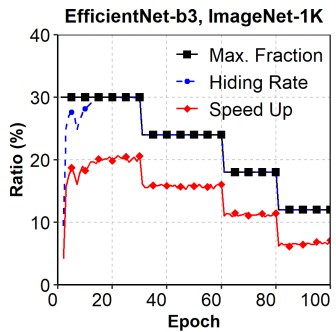

Figure 4: Reduction of hiding fraction, per epoch, and the resulting speedup.

Table 6: The impact of different components of KAKURENBO on testing accuracy including **HE**: Hiding $F\%$ lowest-loss examples, **MB**: Moving Back, **RF**: Reducing the Fraction by epoch, **LR**: Adjusting Learning Rate. Numbers inside the (.) indicate the gap in percentage compared to the full version of KAKURENBO.

|  | Component | | | | Accuracy |
|  | HE | MB | RF | LR |  |
| --- | --- | --- | --- | --- | --- |
| Baseline | × | × | × | × | 73.68 |
| v1000 | ✓ | × | × | × | 72.25 (-1.8%) |
| v1001 | ✓ | × | × | ✓ | 73.08 (-0.7%) |
| v1010 | ✓ | × | ✓ | × | 72.81 (-1.1%) |
| v1011 | ✓ | × | ✓ | ✓ | 73.27 (-0.4%) |
| v1100 | ✓ | ✓ | × | × | 72.37 (-1.7%) |
| v1101 | ✓ | ✓ | × | ✓ | 73.09 (-0.7%) |
| v1110 | ✓ | ✓ | ✓ | × | 72.96 (-0.9%) |
| KAKUR. (v1111) | ✓ | ✓ | ✓ | ✓ | 73.6 |

samples. Thus, KAKURENBO takes more training time in this case. In short, KAKURENBO is optimized for large-scale training and provides more benefits when running on multiple GPUs.

### 4.3 Ablation Studies

**Impact of prediction confidence threshold $\tau$.** Higher prediction confidence threshold $\tau$ leads to a higher number of samples being moved back to the training set, i.e., fewer hidden samples at the beginning of the training process. At the end of the training process, when the model has is well-trained, more samples are predicted correctly with high confidence. Thus the impact of the prediction confidence threshold on the number of moved-back samples becomes less (as shown in Figure 4). The result in Table 5 shows that when we increase the threshold $\tau$, we obtain better accuracy (fewer hidden samples), but at the cost of smaller performance gain. We suggest to set $\tau = 0.7$ in all the experiments as a good trade-off between training time and accuracy.

**Impact of different components of KAKURENBO.** We evaluate how KAKURENBO's individual internal strategies, and their combination, affect the testing accuracy of a neural network. Table 6 reports the results we obtained when training ResNet-50 on ImageNet-1K[3] with a maximum hiding fraction of $40\%$ . The results show that when only HE (Hiding Examples) of the $40\%$ lowest loss samples is performed, accuracy slightly degrades. Combining HE with other strategies, namely MB (Move-Back), RF (Reducing Fraction), and LR (Learning Rate adjustment) gradually improves testing accuracy. In particular, all combinations with RF achieve higher accuracy than the ones without it. For example, the accuracy of v1110 is higher than that of v1100 by about $0.59\%$. We also observe that using LR helps to improve the training accuracy by a significant amount, i.e., from $0.46\%$ to $0.83\%$. The MB strategy also improves accuracy. For example, the accuracy of v1010 is $72.81\%$, compared to v1110 which is $72.96\%$. This small impact of MB on the accuracy is due to moving back samples at the beginning of the training, as seen in Appendix C.3. By using all the strategies, KAKURENBO achieves the best accuracy of $73.6\%$, which is very close to the baseline of $73.68\%$.

## 5 Conclusion

We have proposed KAKURENBO, a mechanism that adaptively hides samples during the training of deep neural networks. It assesses the importance of samples and temporarily removes the ones that would have little effect on the SGD convergence. This reduces the number of samples to process at each epoch without degrading the prediction accuracy. KAKURENBO combines the knowledge of historical prediction confidence with loss and moves back samples to the training set when necessary. It also dynamically adapts the learning rate in order to maintain the convergence pace. We have demonstrated that this approach reduces the training time without significantly degrading the accuracy on large datasets.

---

[3] We use the ResNet-50 (A) configuration in this evaluation as shown in Appendix-B

# 6 Acknowledgments

This work was supported by JSPS KAKENHI under Grant Numbers JP21K17751 and JP22H03600. This paper is based on results obtained from a project, JPNP20006, commissioned by the New Energy and Industrial Technology Development Organization (NEDO). This work was supported by MEXT as "Feasibility studies for the next-generation computing infrastructure" and JST PRESTO Grant Number JPMJPR20MA.
We thank Rio Yokota and Hirokatsu Kataoka for their support on the Fractal-3K dataset.

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

# APPENDIX

## Appendix A. Proof of Lemma 1

**Lemma 1.** *Let $F(\mathbf{w}) = \mathbb{E}[f_i(\mathbf{w})]$ be non-convex. Set $\sigma^2 = \mathbb{E}[\|\nabla f_i(\mathbf{w_M})\|^2]$ with $\mathbf{w}^* := argmin F(\mathbf{w})$. Suppose $\eta \leq \dfrac{1}{\sup_i L_i}$. Let $\Delta_t = \mathbf{w_t} - \mathbf{w}$. After $T$ iterations, SGD satisfies:*

$$\mathbb{E}\left[\|\Delta_T\|^2\right] \leq (1 - 2\eta\hat{C})^T \|\Delta_0\|^2 + \eta R_\sigma$$

*where $\hat{C} = \lambda(1 - \eta \sup_i L_i)$ and $R_\sigma = \dfrac{\sigma^2}{\hat{C}}$.*

*Proof.* $\|\nabla f_i(\mathbf{w})\| = 0$ in the noiseless setting, and so $\sigma := 0$. For $\boldsymbol{x}_k$ being the input at $i$ random index for iteration $k$, there exists a parameter $\lambda_{\mathbf{w_t}}$ for $\lambda_{max}$ (Eq. 7), and $w = w_\lambda$, we have for step size $\gamma$

$$
\begin{aligned}
\mathbb{E}\left[\|\Delta_T\|^2\right] &= \|\boldsymbol{x}_k - \boldsymbol{x}_\star - \gamma\nabla f_i(\boldsymbol{x}_k)\|^2 \\
&= \|(\boldsymbol{x}_k - \boldsymbol{x}_\star) - \gamma(\nabla f_i(\boldsymbol{x}_k) - \nabla f_i(\boldsymbol{x}_\star)) - \gamma\nabla f_i(\boldsymbol{x}_\star)\|^2 \\
&= \|\boldsymbol{x}_k - \boldsymbol{x}_\star\|^2 - 2\gamma\boldsymbol{x}_k - \boldsymbol{x} * \nabla f_i(\boldsymbol{x}_k) + \gamma^2\|\nabla f_i(\boldsymbol{x}_k) - \nabla f_i(\boldsymbol{x}_\star) + \nabla f_i(\boldsymbol{x}_\star)\|^2 \\
&\leq \|\boldsymbol{x}_k - \boldsymbol{x}_\star\|^2 - 2\gamma\boldsymbol{x}_k - \boldsymbol{x} * \nabla f_i(\boldsymbol{x}_k) + 2\gamma^2\|\nabla f_i(\boldsymbol{x}_k) - \nabla f_i(\boldsymbol{x}_\star)\|^2 + 2\gamma^2\|\nabla f_i(\boldsymbol{x}_\star)\|^2 \\
&\leq \|\boldsymbol{x}_k - \boldsymbol{x}_\star\|^2 - 2\gamma\boldsymbol{x}_k - \boldsymbol{x} * \nabla f_i(\boldsymbol{x}_k) \\
&\quad + 2\gamma^2 L_i\boldsymbol{x}_k - \boldsymbol{x}_\star + \nabla f_i(\boldsymbol{x}_k) - \nabla f_i(\boldsymbol{x}_\star) + 2\gamma^2\|\nabla f_i(\boldsymbol{x}_\star)\|^2
\end{aligned}
$$

where we employ Jensen's inequality in the first inequality for $\sigma^2 = \mathbb{E}[\|\nabla f_i(\mathbf{w_M})\|^2]$. Then $\mathbb{E}[\nabla f_i(\boldsymbol{x})] = F(\boldsymbol{x})$, and we obtain

$$
\begin{aligned}
\mathbb{E}\left[\|\Delta_T\|^2\right] &\leq \|\boldsymbol{x}_k - \boldsymbol{x}_\star\|^2 - 2\gamma\boldsymbol{x}_k - \boldsymbol{x}_\star * F(\boldsymbol{x}_k) + 2\gamma^2\mathbb{E}\left[L_i\boldsymbol{x}_k - \boldsymbol{x}_\star, \nabla f_i(\boldsymbol{x}_k) - \nabla f_i(\boldsymbol{x}_\star)\right] \\
&\quad + 2\gamma^2\mathbb{E}\|\nabla f_i(\boldsymbol{x}_\star)\|^2 \\
&\leq \|\boldsymbol{x}_k - \boldsymbol{x}_\star\|^2 - 2\gamma\boldsymbol{x}_k - \boldsymbol{x}_\star * F(\boldsymbol{x}_k) + 2\gamma^2 \sup_i L_i\mathbb{E}\boldsymbol{x}_k - \boldsymbol{x}_\star, \nabla f_i(\boldsymbol{x}_k) - \nabla f_i(\boldsymbol{x}_\star) \\
&\quad + 2\gamma^2\mathbb{E}\|\nabla f_i(\boldsymbol{x}_\star)\|^2 \\
&= \|\boldsymbol{x}_k - \boldsymbol{x}_\star\|^2 - 2\gamma\boldsymbol{x}_k - \boldsymbol{x}_\star * F(\boldsymbol{x}_k) + 2\gamma^2 \sup L\boldsymbol{x}_k - \boldsymbol{x}_\star, F(\boldsymbol{x}_k) - F(\boldsymbol{x}_\star) + 2\gamma^2\sigma^2
\end{aligned}
$$

when $\gamma \leq \frac{1}{\sup L}$. Recursively applying this bound over the first $k$ iterations yields the desired result

$$
\begin{aligned}
\mathbb{E}\left[\|\Delta_T\|^2\right] &\leq \left(1 - 2\gamma\mu(1 - \gamma)\right)^k \|\boldsymbol{x}_0 - \boldsymbol{x}_\star\|^2 + 2\sum_{j=0}^{k-1}\left(1 - 2\gamma\mu(1 - \gamma)\right)^j \gamma^2\sigma^2 \\
&\leq \left(1 - 2\gamma\mu(1 - \gamma)\right)^k \|\boldsymbol{x}_0 - \boldsymbol{x}_\star\|^2 + \frac{\gamma\sigma^2}{\mu(1 - \gamma)}.
\end{aligned}
$$

$\square$

## Appendix B. Experiments Details

### B.1. System detail

We run our experiments on a supercomputer with 1000s of compute nodes, each equipped with 2 Intel Xeon Gold 6148 CPUs, 384 GiB of RAM, 4 NVidia V100 GPUs, and Infiniband EDR NICs (100Gbps×2). We run 4 MPI ranks per compute node so that each rank has a dedicated access to a GPU.

Table 7: Datasets and Models Used in Experiments (* Down-stream training using the pre-trained model).

| Model | Dataset | #Samples | #Epoch | #GPUs | minibatch (per GPU) | Task |
|---|---|---|---|---|---|---|
| Resnet50 [36] | ImageNet-1K [19] | 1.2M | 100 | 32 | 64 | Image Classification |
| EfficientNet-b3 [37] | | | | | 32 | |
| WideResNet-28-10 [39] | CIFAR-100 [41] | 50K | 200 | 32 | 32 | Image Classification |
| DeepCAM [4] | DeepCAM [4] | ∼ 122K | 35 | 1024 | 1 | Image Segmentation |
| DeiT-Tiny-224 [42] | Fractal-3K [6] | 3M | 80 | 32 | 16 | Image Classification |
| | (*) CIFAR-10 [41] | 50K | 1000 | 8 | 96 | |
| | (*) CIFAR-100 [41] | 50K | 1000 | 8 | 96 | |

## B.2. Model training method details and dataset information:

Table 7 summarizes the models and datasets used in this work. In details, we evaluate KAKURENBO using several models on various datasets as the following:

- **ImageNet-1K** [19]: We use the subset of the ImageNet dataset containing 1000 classes each containing around 1300 images (1,282,048 images in total). We also test the trained model on the validation set of $50,000$ samples. We train ResNET-50 and EfficientNet-b3 provided by 'torchvision v0.12.0' on ImageNet-1K dataset.

- **CIFAR-10/CIFAR-100** [41]: The CIFAR-10/CIFAR-100 dataset dataset consists of 60,000 colour images. It has 100 categories each containing 600 images. The dataset provides 50,000 training images and 10,000 test images with a size of $32 \times 32$ pixels. CIFAR-100 dataset is available at https://www.cs.toronto.edu/ kriz/cifar.html.

- **DeepCAM** [4]: DeepCAM dataset for image segmentation, which consists of approximately 122K samples and requires 8.8TBs of storage. We use the settings in [4] to train DeepCAM with the top learning rate of $0.0055$.

- **Fractal-3K** [6] A rendered dataset from the Visual Atom method [6]. Fractal-3K dataset comprise of 3 million images of visual atoms, where the number of classes is C = 3000 and the number of images per class is N = 1000. We train the DeiT-Tiny-224 model on Fractal-3K dataset and fine tune it with CIFAR-10 and CIFAR-100 datasets.

## B.3. Hyper-parameters

It is worth noting that we follow the hyper-parameters reported in [38] for training ResNet-50, [39] for training WideResNet-28-10, [37] for training EfficientNet-b3, and [4] for DeepCAM. We also use the setting in [6] for both pretrain and finetune tasks in Fractal-3K. Table 8 shows the detail of our hyper-parameters. Specifically, We follow the guideline of 'TorchVision' to train the ResNet-50 that uses the CosineLR learning rate scheduler [4], auto augments, and random erasing, etc [38]. We also set the weight decay to $1e - 05$ and crop the input image to $176 \times 176$ pixels and train for a long number of epochs, i.e., 600 (The ResNet-50 setting). We train the WideResNet-28-4 on the CIFAR-100 dataset in 200 epochs following the setting in [39]. Specifically, we use the base learning rate of $0.025 \times k$, momentum 0.9, and weight decay 0.0005. For EfficientNet-b3, we use RMSProp optimizer with momentum 0.9; batch norm momentum 0.99 weight decay $1e - 5$ (following [37]). We use an initial learning rate of $0.016$ that decays by 0.9 every 2 epochs. We set the minibatch size

---

[4]implemented by timm https://github.com/huggingface/pytorch-image-models/tree/main/timm

Table 8: Hyper-parameters used for different training in the paper and the baseline top-1 testing accuracy. We also considers different hyper-parameters for ResNet-50 model on ImageNet-1K dataset.

| | ImageNet-1K | | | | CIFAR-100 | Fractal-3K | CIFAR-10 | CIFAR-100 |
|---|---|---|---|---|---|---|---|---|
| | ResNet-50 | ResNet-50 (A) | ResNet-50 (B) | EfficientNet-b3 | WideResNet-28-10 | DeiT-Tiny-224 | | |
| Train Res | 224 | 224 | 224 | 224 | 32 | 224 | 224 | 224 |
| Test Res | 232 | 224 | 232 | 224 | 32 | - | 32 | 32 |
| Epochs | 600 | 100 | 600 | 100 | 200 | 80 | 1000 | 1000 |
| Number of workers | 32 | 32 | 32 | 32 | 32 | 32 | 8 | 8 |
| Batch size | 2048 | 1024 | 1024 | 1024 | 1024 | 512 | 768 | 768 |
| Optimizer | SGD | SGD | SGD | SRMSProp | SGD | adamw | SGD | SGD |
| Momentum | 0.9 | 0.9 | 0.9 | 0.9 | 0.9 | - | 0.9 | 0.9 |
| LR | 0.11 | 0.0125 | 0.125 | 0.01 | 0.025 | 0.001 | 0.01 | 0.01 |
| Weight decay | 1e-5 | 5e-5 | 2e-5 | 5e-5 | 5e-4 | 0.05 | 1e-4 | 1e-4 |
| LR decay | cosineLR | step | cosineAnnealing | step | step | Cosine_iter | Cosine_iter | Cosine_iter |
| Decay rate | - | 0.1 | - | 0.9 | 0.2 | - | - | - |
| Decay epochs | - | [30, 60, 80] | - | 2 | [60, 120, 160] | - | - | - |
| Warmup epochs | 5 | 5 | 5 | 5 | 1 | 5 | 5 | 5 |
| Warmup method | linear | linear | linear | linear | linear | linear | linear | linear |
| Label Smoothing | 0.1 | - | - | - | - | - | 0.1 | 0.1 |
| H.flip | YES | YES | YES | YES | YES | YES | YES | YES |
| Erasing prob. | 0.1 | - | 0.1 | - | - | 0.5 | 0.5 | 0.5 |
| Auto augment | ta_wide | - | ta_wide | - | - | rand-m9-mstd0.5-inc1 | | |
| Interpilation | bilinear | - | bilinear | - | - | bicubic | bicubic | bicubic |
| Train crop | 176 | - | 176 | - | - | 224 | 224 | 224 |
| Test crop | 224 | - | 224 | - | - | - | - | - |
| EMA | YES | - | - | - | - | - | - | - |
| EMA steps | 32 | - | - | - | - | - | - | - |
| EMA decay | 0.99998 | - | - | - | - | - | - | - |
| Loss | Cross Entropy | Cross Entropy | Cross Entropy | Cross Entropy | Cross Entropy | Cross Entropy | Soft Target Cross Entropy | |
| **Baseline acc.** | 74.89 | 73.68 | 76.58 | 76.63 | 77.49 | - | 95.03 | 79.69 |
| Max fraction | 0.3 | 0.3 | 0.3 | 0.3 | 0.3 | 0.3 | - | - |
| Max fraction decay | [1, 0.8, 0.6] | [1, 0.8, 0.6] | [1, 0.8, 0.6] | [1, 0.8, 0.6] | [1, 0.8, 0.6] | [1, 0.8, 0.6] | - | - |
| Fraction decay epoch | [200, 400, 600] | [30, 60, 80] | [200, 400, 600] | [30, 60, 80] | [60, 120, 160] | [30, 60, 80] | - | - |
| **KAKURENBO acc.** | 75.15 | 73.52 | 76.62 | 76.23 | 77.21 | - | 95.28 | 79.35 |

per worker (GPU) to $b$, e.g., the global batch size of $b \times p$ in the case of $p$ GPUs. The minibatch size per GPU and the number of GPUs in each experiments are shown in Table 7.

To show the robustness of KAKURENBO, we also train ResNet-50 with different settings, e.g., marked as (A) and (B) in the Table 8 and discuss the result in Appendix C.3. For example, in ResNet-50 (A) setting, we follow the hyper-parameters reported in [34]. Specifically, we use using the Stochastic Gradient Descent (SGD) optimizer with a Nesterov momentum of $0.9$ and weight decay of $0.00005$. We trained all the models for 100 epochs and apply the linear scaling rule with the base learning rate of $0.0125 \times k$ where $k$ is the number of workers. We reduce the learning rate by $0.1$ at the 30th, 60th, and 80th epoch. We gradual warmup which starts with $0$ and is linearly increased to the base learning rate over $5$ epochs. We also use scale and aspect ratio data augmentation.The input image is a $224 \times 224$ pixel random crop from an augmented image or its horizontal flip.

## B.4. Implementation

It is worth noting that KAKURENBO merely hides samples before the input pipeline. As a result, KAKURENBO can be easily implemented with simple extensions to PyTorch and TensorFlow implementations[5]. Using KAKURENBO with new models and datasets can be added to any training code by indicating so in the model launch parameters.

---

[5] Our PyTorch implementation is available at https://github.com/TruongThaoNguyen/kakurenbo

# Appendix C. Ablation Studies

### C.1. Analysis of the Factors Affecting KAKURENBO's Performance

In this section, we present an analysis of the factors affecting KAKURENBO's performance, e.g., the lagging loss and the prediction confidence.

**The loss.** Figure 5 shows the histogram of the loss as the number of epochs increases when training ResNet-50 (A) on the ImageNet-1K dataset. At the first few epochs, the histogram of the loss follows a Gaussian distribution. As the number of epochs increases, the number of samples with small loss increases significantly. For example, starting from epoch 30, more than $50\%$ of the samples have a loss which is lower than 5% of the highest loss. As a result, there is an increase in the number of samples that provide about the same absolute contribution to the update, e.g., in the latter epochs. Hiding a fraction of (fixed) $F$ samples during training in this case may lead to a relatively higher negative impact on the accuracy than that at some early epochs. Thus, we reduce the maximum hidden fraction at the epoch number increases (as mentioned in Section 3.4 in the main manuscript).

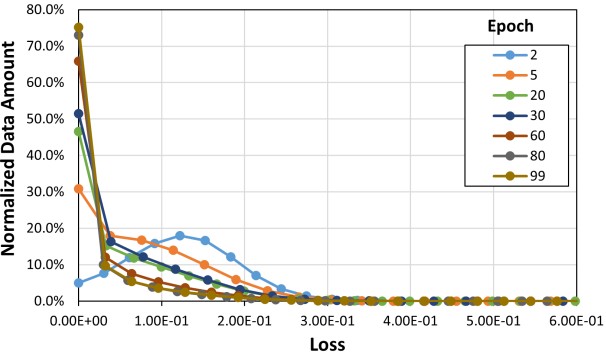

Figure 5: Histogram of the lagging-loss as the number of epoch increases during training (ResNet-50 w/ ImageNet-1K).

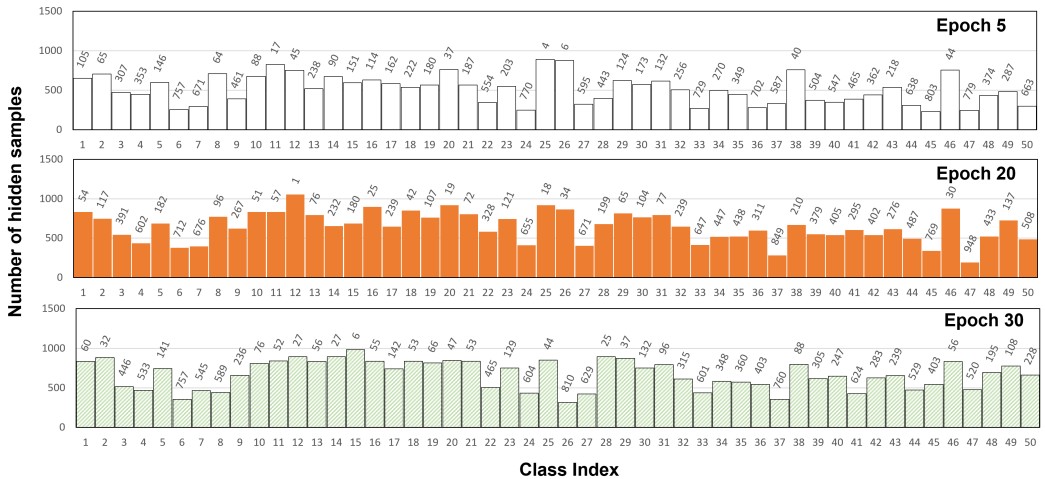

Figure 6: Number of hidden samples of each class in KAKURENBO (ResNet-50, ImageNet-1K). The figure shows the result of the first 50 classes. The number on top of each column shows the rank over 1000 classes (a lower rank indicates a higher number of hidden samples).

In addition, as the number of samples with the same absolute loss increases, there is a high probability that samples classified as important are in fact unimportant. To this end, we propose moving back samples from the hidden set based on their prediction confidence score (as per Section 3.2). Unlike the ahead-of-time method proposed by the authors in [13], instead of computing the loss of all the

samples before training and selecting samples to be removed from the training process, we compute the loss of the samples on the fly. With this method, at each epoch, a dynamic hiding fraction $F^*$ is applied. Figure 6 shows the number of hidden samples of each class in KAKURENBO (ResNet-50, ImageNet-1K). The figure shows the result of the first 50 classes. The number on top of each column shows the rank over 1000 classes (a lower rank indicates a higher number of hidden samples). The result shows that our method could dynamically hide the samples at each epoch. For example, fewer samples in the class 25 are hidden while more and more samples in class 13 are selected to hide as epochs increase. Figure 7 shows that the impact of each class remains different during the training. Easy classes such as class 1, and 2 are hidden during the training more than the class 31 and 47. The result in Figure 8 also shows that in two continuous epochs, only (around) 30% of samples are hidden again. This result infers that the importance (or contribution level) of samples is changing epoch by epoch.

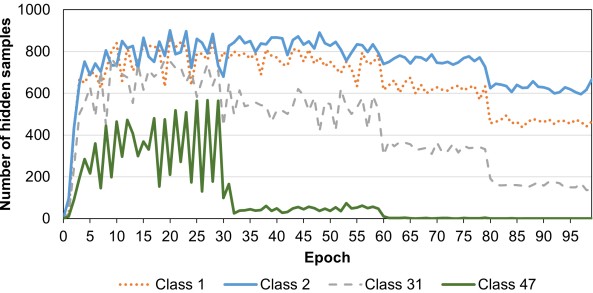

Figure 7: Number of hidden samples of each class in KAKURENBO (ResNet-50, ImageNet-1K). The figure shows the result of the first selected classes for a better presentation. The impact of each class (or each sample) is different in training. Easy classes such as class 1, 2 are hidden during the training more than the class 31 and 47.

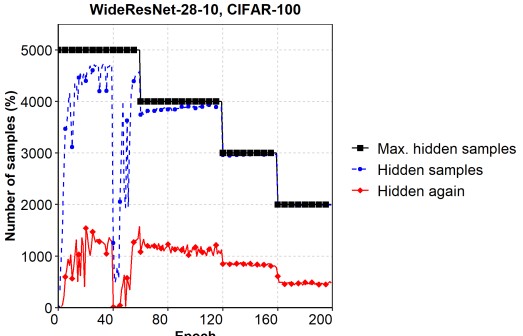

Figure 8: Number of hidden samples per epoch. **Max. hidden samples** presents the number of samples considered for hidding in each epoch, e.g., which is in proportional to the fraction $F$. **Hidden samples** is the actual fraction of hidden samples in each epoch (after moving samples from the hidden list back to the training list). **Hidden again** presents the number of samples that were hidden in an epoch $i$ and also hidden in the epoch $i-1$. In general, only around 30% of the samples are hidden again in each epoch. The number of samples that are moved back becomes smaller when the epoch increases because the prediction confidence becomes higher during the training.

**The calibration of the softmax prediction confidence** As mentioned in the main manuscript, only samples that have low loss and sustain correct prediction with high confidence in the current epoch are hidden in the following epoch. A sample is correctly predicted with high confidence at an epoch $e$ if it is predicted correctly (**PA**) and the prediction confidence (**PC**) is no less than a certain threshold $\tau$. As shown in Figure 8, the number of samples moved back becomes smaller when as the number of epochs increases because the prediction confidence becomes higher during the training, thus the actual hidden samples become similar to the max. hidden samples in the latter epoch. Results in

Table 9: Max testing accuracy (Top-1) of KAKURENBO in the comparison with Baseline. **Reported Acc.** represent the accuracy reported in the main manuscript. **New Acc.** represent the accuracy achieved in 3 different runs with different random seeds.

| Setting | CIFAR-100 WRN-28-10 | | ImageNet-1K ResNet-50(A) | |
|---|---|---|---|---|
| | **Reported Acc.** | **New Acc.** | **Reported Acc.** | **New Acc.** |
| Baseline | 77.49 | $77.26 \pm 0.55$ | 73.68 | $74.06 \pm 0.08$ |
| KAKURENBO | 77.21 | $77.20 \pm 0.63$ | 73.52 | $73.65 \pm 0.07$ |
| Random | - | $76.82 \pm 0.43$ | - | - |

Table 10: Test accuracy (Top-1) in percentage and total training time in seconds of KAKURENBO in the comparison with those of the baseline.

| Setting | ResNet-50(A) + ImageNet-1K | | ResNet-50(B) + ImageNet-1K | |
|---|---|---|---|---|
| | **Accuracy** | **Time** (sec) | **Accuracy** | **Time** (sec) |
| Baseline | 73.68 | 16118 | 76.58 | 64060 |
| KAKURENBO-0.2 | - | - | 76.11 | 61723 |
| KAKURENBO-0.3 | 73.52 | 12984 | 76.17 | 59063 |
| KAKURENBO-0.4 | - | - | 75.62 | 57582 |

Table 5 show that when we increase the threshold $\tau$ we obtain better accuracy (i.e., fewer hidden samples) at the cost of smaller performance improvements. However, the gaps remain small.

## C.2. Evolution of the Hiding Fraction

Figure 4 shows how KAKURENBO adapts the size of the hidden set during the training of EfficientNet-b3. At the beginning of the training, the maximum hiding fraction is set to 30 %. This fraction is progressively reduced after a few epochs followed by our fraction adjustment rule. The figure also reports the effective proportion of samples that are hidden at each epoch (*Hiding rate* in the Figure). As described in Section 3 in the main manuscript, KAKURENBO first cuts a part of the dataset before moving back samples that are mispredicted or correctly predicted but with low confidence. Figure 4 shows that the moving back strategy mostly impacts the beginning of the training when the model is still inaccurate.

Figure 4 also reports the measured speedup per epoch as compared to the baseline epoch duration. The speedup follows the same trend as the hiding rate. This is because reducing the number of samples in the training set impacts the speed of the training. The measured speedup does not reach the maximum hiding rate because of additional hidden sample selection and due to the need for computing the forward pass on samples in the hidden list.

## C.3. Robustness of Our Method

In this section, we first confirm that the results of our implementation are stable by running experiments multiple times. Due to resource limitations, we could not run experiments multiple times for each data point reported in the paper, especially for models and datasets that are big in this work. The result for CIFAR-100 and ImangetNet-1K is reported in Table 9.The result shows that KAKURENBO is stable with different random seeds.

We now demonstrate the robustness of KAKURENBO with different settings during training, e.g. (1) when using different techniques to improve accuracy and (2) the batch size is changed.

We first measure the robustness of KAKURENBO when using SoTA techniques in training, the ResNet-50 (A) and (B) described in Table 8. The result in Figure 9 and Table 10 show that our proposed method is also stable with different learning techniques. For example, KAKURENBO could reduce the total training time to $19.5\%$ ($7.8\%$) with only $0.2\%$ ($0.41$) percent of accuracy reduction when the maximum hidden fraction is set to $30\%$ for RESNET-50 (A) and (B), respectively.

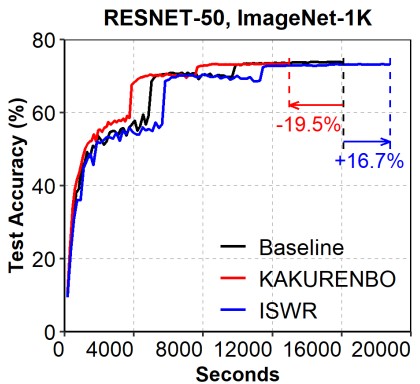 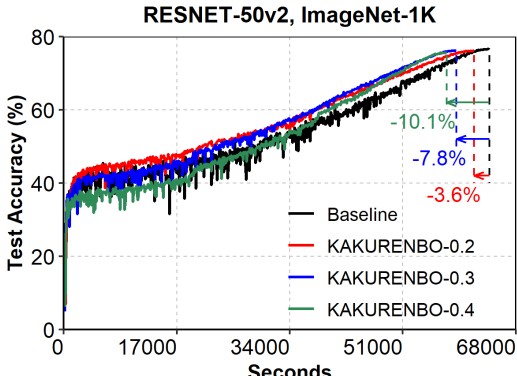

Figure 9: Convergence and speedup of KAKURENBO with different settings of ResNet-50 including [LEFT] ResNet-50 (A) and [RIGHT] ResNet-50 (B).

Table 11: Test accuracy (Top-1) in percentage of KAKURENBO in comparison with those of the baseline when the batch size changes.

| Setting | ResNet-50 (A) + ImageNet-1K | | | |
|---|---|---|---|---|
| #GPUs | 32 | 64 | 128 | 256 |
| Batch size | 1024 | 2048 | 4096 | 8192 |
| Baseline | 73.68 | 73.98 | 73.59 | 73.81 |
| KAKURENBO-0.4 | 73.60 | 73.21 | 73.03 | 72.84 |

We now fix the mini-batch size per worker to 32 and then increase the number of workers (GPUs), i.e., we increase the global batch size in the case of ResNet-50 (A). Table 11 shows the top-1 testing accuracy of ResNet-50 (A) on the ImageNet-1K dataset when the batch size changes from 1024 to 8192. The result shows that KAKURENBO can maintain the accuracy (or with a trivial reduction of accuracy) even with large batch sizes. KAKURENBO could help with large-scale training which has become common when training DL models on a large supercomputer or cluster.

### C.4. Comparison with other methods

We provide extra results in Table 9 to evaluate the training accuracy of random hiding with the CIFAR-100 dataset and WRN-28-10 model. As seen, accuracy is only 76.82% which is lower than that of both KAKURENBO and Baseline. In fact, randomly hiding samples has been investigated before in the GradMatch paper and it's been reported that accuracy is low. This drove us originally not to evaluate this method.

It is important to note that for method [28], the reported speedup is for a specific training regime (that uses a particular optimizer: LARS). The specific training regime described in the paper leads to a slow baseline (see Table 2 in [28], which shows the baseline that trains ImageNet-1K/ResNet-50 on 8 A100 GPUs for 90 epochs to be 3-4x slower than the typical number of hours to train ImageNet-1K/ResNet-50 on 8 A100 GPUs, as reported by many sources, including Nvidia NGC catalog[6]). That means while InfoBatch reports 26% in speedup, the baseline setting for which InfoBatch is demonstrated to be effective is slow.

Basically the proposed method in [43] has a high overhead of visiting each sample to find the coreset. In addition, it is only demonstrated on small datasets such as CIFAR and TinyImageNet, at which the overheads for a large number of samples would not appear. KAKURENBO on the other hand is demonsrated on ImageNet-1K, DeepCAM, and ImageNet-1K-Fractal.

---

[6]https://catalog.ngc.nvidia.com/orgs/nvidia/teams/dle/resources/resnet_pyt/performance

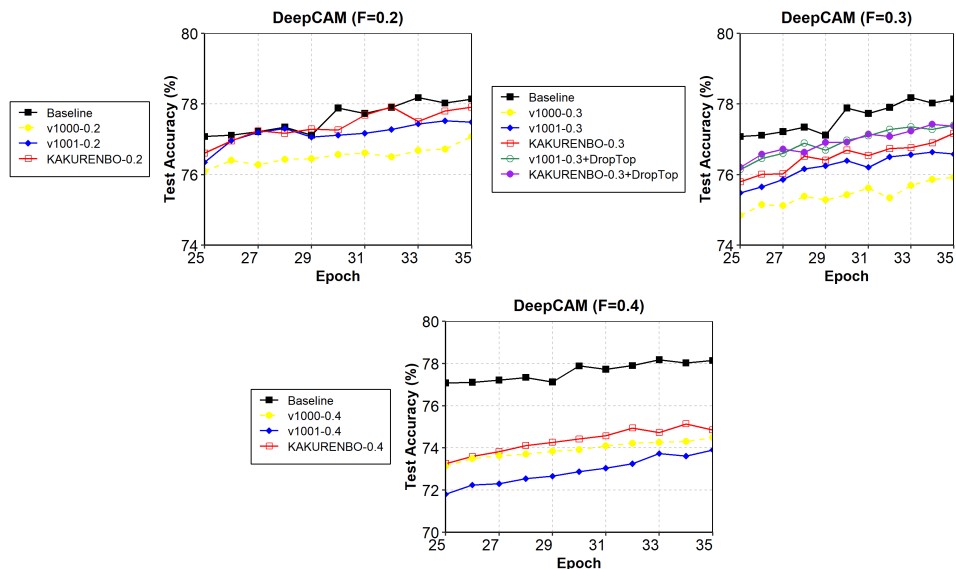

Figure 10: The impact of different components of KAKURENBO on testing accuracy (DeepCAM). **v1000**: Hiding $F$% lowest-loss samples only (HE). **v1001**: HE + LR (Adjusting Learning Rate). **KAKURENBO**: our proposed method with HE + LR + MB (Moving Back) + FR (Reducing the Fraction by epoch). We also consider the version in which we cut 2% of the highest-loss samples at each epoch (DropTop).

## Appendix D. Discussion on DeepCAM

We have shown how KAKURENBO's internal strategies, and their combination, affect the testing accuracy of a neural network in the case of ResNet-50 and the ImageNet-1K dataset. Figure 10 presents the same result on the DeepCAM dataset. In this experiment, we evaluate two combinations: **v1000** and **v1001**. For **v1000** we hide $F$% lowest-loss samples only (Hiding Example or HE for short). For **v1001** we combine HE and learning rate adjustment techniques. It is worth noting that our proposed method, **KAKURENBO**, is the combination of HE, LR, MB (Moving Back sample), and FR (Reducing the Fraction by epoch). The result with different maximum hidden fractions, e.g. $F$ from 0.2 to 0.4, shows that using LR helps to improve the training accuracy by a significant amount, and KAKURENBO achieves the best accuracy which is very close to the baseline. This result is similar to what we observed with ResNet-50 and the ImageNet-1K dataset.

For DeepCAM, we also observed that the loss of the samples with the highest loss does not decrease significantly during the last few epochs of training and remain substantially above the rest of other samples. Those samples may be hard to learn or represent noise in the data. Figure 11 demonstrates this phenomena showing the loss distributions of the full, bottom 98% and top 2% of the dataset according to the loss values, respectively. As seen, the top 2%'s loss distribution remains high until the very last epoch.

This observation motivated us to consider a version in which we cut 2% of the highest-loss samples at each epoch (DropTop). Interestingly, it helps to improve the testing accuracy of DeepCAM, e.g., from 77.16% in KAKURENBO to 77.37% with a maximum fraction of 0.3. For version v1001, Droptop increases the accuracy by 0.82%.

## Appendix E. Related work

As the size of training datasets and the complexity of deep-learning models increase, the cost of training neural networks becomes prohibitive. Several approaches have been proposed to reduce this training cost without degrading accuracy significantly.

**Biased with-Replacement Sampling** has been proposed as a method to improve the convergence rate in SGD training [11, 12]. Importance sampling is based on the observation that not all samples are of

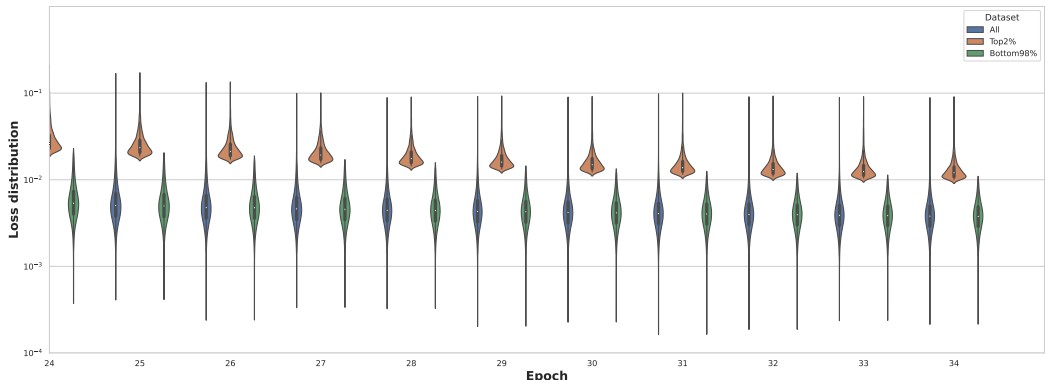

Figure 11: Loss distributions of DeepCAM training samples (full dataset, bottom 98% and top 2%) in the last 10 epoch of training.

equal *importance* when it comes to training, and accordingly replaces the regular uniform sampling used to draw samples from datasets with a biased sampling function that assigns a likelihood to a sample being drawn proportional to its importance; the more important the sample is, the higher the likelihood it would be selected. The with-replacement strategy of importance sampling maintains the total number of samples the network trains on.

Several improvements over importance sampling have been proposed. Reducible Holdout Loss Selection (RHO-LOSS) [12] is a selection function that quantifies by how much each sample would reduce the loss on unseen data had it been trained on. Mercury uses an importance-aware data sharding technique in order to speed up distributed training [22]. It distributes important samples across workers between iterations. This allows important samples to be uniformly distributed between workers, and it reduces the number of samples to communicate for each epoch since non-important samples are kept local.

The importance of a sample can be estimated with several methods. In [23], authors use distance weighted sampling to determine the importance of samples. [24] uses stochastic optimization to reduce the stochastic variance. [25] selects each coordinate with a probability proportional to the square root of its smoothness parameter (applied to accelerated coordinate descent). RAIS [26] proposes approximating the ideal sampling distribution, which introduces little computational overhead.

Overall, biased with-replacement sampling aims at increasing the convergence speed of SGD by focusing on samples that induce a measurable change in the model parameters, which would allow a reduction in the number of epochs. While these techniques promise to converge in fewer epochs on the whole dataset, each epoch requires computing the importance of samples which is time-consuming; and the actual speedup in terms of time-to-solution remains unclear. Moreover, existing studies [11, 12, 22] only evaluate small datasets. Our experiments show that the biased with-replacement, importance sampling [11], the algorithm does not speedup the training when applied to large-scale datasets (demonstrated in the evaluation section in the paper).

**Data Pruning techniques** are used to reduce the size of the dataset by removing less important samples. Pruning the dataset requires training on the full dataset and adds significant overheads for quantifying individual differences between data points [27]. However, the assumption is that the advantage would be a reduced dataset that replaces the original datasets when used by others to train. Several studies investigate the selection of the samples to discard from a dataset. In [13], authors detect unforgettable samples that are correctly classified during the course of training. EL2N [15] uses the loss gradient norm of samples to identify the important ones and prune the unimportant samples from the dataset after a few epochs. While this work does not require fully training the model before pruning, it remains unclear if EL2N reduces the total training time. Another work uses memorization to identify outliers or mislabeled samples in a given dataset [14]. Removing these atypical samples accelerates the training without altering the trained model accuracy. Ensemble Active Learning [16] trains an ensemble of networks and uses ensemble uncertainty to identify which samples are hard to learn. They manage to reduce the ImageNet dataset by 20% without degrading

the accuracy of the trained model, but again, their method is prohibitive for models and datasets that require excessive resources for training.

Pruning the dataset does reduce the training time without significantly degrading the accuracy [13, 14]. However, these techniques require fully training the model on the whole dataset to identify the samples to be removed, which is compute intensive. While most of the proposed solutions perform well on small datasets such as CIFAR, many fail to maintain accuracy on larger datasets like ImageNet [27].

**Selective-Backprop** [17] combines importance sampling and online data pruning. It reduces the number of samples to train on by using the output of each sample's forward pass to estimate the sample's importance and cuts a fixed fraction of the dataset at each epoch. While this method shows notable speedups, it has been evaluated only on tiny datasets without providing any measurements on how accuracy is impacted. In addition, the authors allow up to 10% reduction in test error in their experiments. EIF [44] is similar to Selective-Backprop: it reduces the computation cost of training by filtering out the samples with the lowest loss. $E^2$-Train [45] shows that the combination of randomly dropping samples during training with selective layer update in CNNs can significantly reduce the training time, while slightly degrading the accuracy. However, $E^2$-Train targets edge environments and is evaluated only on very small datasets.

**GRAD-MATCH** [18] is an online method that selects a subset of the samples that would minimize the gradient matching error, where the error of the gradients of a matched subset samples (and their weights) becomes minimum. To avoid the impractical storing and computation of the optimization of the gradients of all instances, the authors approximate the gradients by only using the gradients of the last layer, use a per-class approximation, and run data selection every $R$ epochs, in which case, the same subsets and weights will be used between epochs. The infrequent selection, however, means the model is limited in its capacity to learn in intermediate epochs - where selection occurs - since it trains on the same limited subset of samples. This often leads to a longer numbers of epochs needed to converge to the same validation accuracy that can be achieved by the baseline or the baseline reaching much higher accuracy [29]. Another important point worth mentioning is that GRAD-MATCH is impractical in distributed training, which is a de facto requirement in large dataset and models (e.g., the DeepCAM model/dataset). That is since the approximation of the classes would require very expensive high-volume collective communication operations to gather the gradients scattered across different samples belonging to the same class. The communication cost would be O($N.R.G$) where $N$ is the number of samples, $R$ is the frequency of selection, and $G$ is the gradients (of the last layer, if gradient approximation is to be used). Distributed GRAD-MATCH would require a scatter communication to collect the class approximations and a collective all-reduce of the gradients to then do the matching optimization. This is practically a very high cost for communication per epoch that could even exceed the average time per epoch. Finally, the mini-batch variant of GRAD-MATCH can only be effective for small mini-batches. However, since in distributed training the mini-batch grows with the scale (i.e., the mini-batch aggregates the local mini-batch of all workers), the cost of communication amplifies by B (where B is mini-batch size).

