# Appendix

# [KAKURENBO: Adaptively Hiding Samples in Deep Neural Network Training]

## 1 Appendix A. Proof of Lemma 1

2 **Lemma 1.** *Let $F(\mathbf{w}) = \mathbb{E}[f_i(\mathbf{w})]$ be non-convex. Set $\sigma^2 = \mathbb{E}[\|\nabla f_i(\mathbf{w_M})\|^2]$ with $\mathbf{w}^* :=$*

3 *$argminF(\mathbf{w})$. Suppose $\eta \leq \dfrac{1}{\sup_i L_i}$. Let $\Delta_t = \mathbf{w_t} - \mathbf{w}$. After $T$ iterations, SGD satisfies:*

$$\mathbb{E}\left[\|\Delta_T\|^2\right] \leq (1 - 2\eta\hat{C})^T \|\Delta_0\|^2 + \eta R_\sigma$$

4 *where $\hat{C} = \lambda(1 - \eta \sup_i L_i)$ and $R_\sigma = \dfrac{\sigma^2}{\hat{C}}$.*

5 *Proof.* $\|\nabla f_i(\mathbf{w})\| = 0$ in the noiseless setting, and so $\sigma := 0$. For $\boldsymbol{x}_k$ being the input at $i$ random
6 index for iteration $k$, there exists a parameter $\lambda_{\mathbf{w_t}}$ for $\lambda_{max}$ (Eq. 7), and $w = w_\lambda$, we have for step
7 size $\gamma$

$$
\begin{aligned}
\mathbb{E}\left[\|\Delta_T\|^2\right] &= \|\boldsymbol{x}_k - \boldsymbol{x}_\star - \gamma \nabla f_i(\boldsymbol{x}_k)\|^2 \\
&= \|(\boldsymbol{x}_k - \boldsymbol{x}_\star) - \gamma(\nabla f_i(\boldsymbol{x}_k) - \nabla f_i(\boldsymbol{x}_\star)) - \gamma \nabla f_i(\boldsymbol{x}_\star)\|^2 \\
&= \|\boldsymbol{x}_k - \boldsymbol{x}_\star\|^2 - 2\gamma \boldsymbol{x}_k - \boldsymbol{x}_\star * \nabla f_i(\boldsymbol{x}_k) + \gamma^2 \|\nabla f_i(\boldsymbol{x}_k) - \nabla f_i(\boldsymbol{x}_\star) + \nabla f_i(\boldsymbol{x}_\star)\|^2 \\
&\leq \|\boldsymbol{x}_k - \boldsymbol{x}_\star\|^2 - 2\gamma \boldsymbol{x}_k - \boldsymbol{x}_\star * \nabla f_i(\boldsymbol{x}_k) + 2\gamma^2 \|\nabla f_i(\boldsymbol{x}_k) - \nabla f_i(\boldsymbol{x}_\star)\|^2 + 2\gamma^2 \|\nabla f_i(\boldsymbol{x}_\star)\|^2 \\
&\leq \|\boldsymbol{x}_k - \boldsymbol{x}_\star\|^2 - 2\gamma \boldsymbol{x}_k - \boldsymbol{x}_\star * \nabla f_i(\boldsymbol{x}_k) \\
&\quad + 2\gamma^2 L_i \boldsymbol{x}_k - \boldsymbol{x}_\star + \nabla f_i(\boldsymbol{x}_k) - \nabla f_i(\boldsymbol{x}_\star) + 2\gamma^2 \|\nabla f_i(\boldsymbol{x}_\star)\|^2
\end{aligned}
$$

8 where we employ Jensen's inequality in the first inequality for $\sigma^2 = \mathbb{E}[\|\nabla f_i(\mathbf{w_M})\|^2]$. Then
9 $\nabla f_i(\boldsymbol{x}) = F(\boldsymbol{x})$, and we obtain

$$
\begin{aligned}
\mathbb{E}\left[\|\Delta_T\|^2\right] &\leq \|\boldsymbol{x}_k - \boldsymbol{x}_\star\|^2 - 2\gamma \boldsymbol{x}_k - \boldsymbol{x}_\star * F(\boldsymbol{x}_k) + 2\gamma^2 \left[L_i \boldsymbol{x}_k - \boldsymbol{x}_\star, \nabla f_i(\boldsymbol{x}_k) - \nabla f_i(\boldsymbol{x}_\star)\right] \\
&\quad + 2\gamma^2 \|\nabla f_i(\boldsymbol{x}_\star)\|^2 \\
&\leq \|\boldsymbol{x}_k - \boldsymbol{x}_\star\|^2 - 2\gamma \boldsymbol{x}_k - \boldsymbol{x}_\star * F(\boldsymbol{x}_k) + 2\gamma^2 \sup_i L_i \boldsymbol{x}_k - \boldsymbol{x}_\star, \nabla f_i(\boldsymbol{x}_k) - \nabla f_i(\boldsymbol{x}_\star) \\
&\quad + 2\gamma^2 \|\nabla f_i(\boldsymbol{x}_\star)\|^2 \\
&= \|\boldsymbol{x}_k - \boldsymbol{x}_\star\|^2 - 2\gamma \boldsymbol{x}_k - \boldsymbol{x}_\star * F(\boldsymbol{x}_k) + 2\gamma^2 \sup L \boldsymbol{x}_k - \boldsymbol{x}_\star, F(\boldsymbol{x}_k) - F(\boldsymbol{x}_\star) + 2\gamma^2 \sigma^2
\end{aligned}
$$

Submitted to 37th Conference on Neural Information Processing Systems (NeurIPS 2023). Do not distribute.

Table 1: Datasets and Models Used in Experiments (* Down-stream training using the pre-trained model).

| Model | Dataset | #Samples | #Epoch | #GPUs | minibatch (per GPU) | Task |
|---|---|---|---|---|---|---|
| Resnet50 He u. a. (2016) | ImageNet-1K Deng u. a. (2009) | 1.2M | 100 | 32 | 64 | Image Classification |
| EfficientNet-b3 Tan und Le (2019) | | | | | 32 | |
| WideResNet-28-10 Zagoruyko und Komodakis (2016) | CIFAR-100 Krizhevsky und Hinton (2009) | 50K | 200 | 32 | 32 | Image Classification |
| DeepCAM Kurth u. a. (2018) | DeepCAM Kurth u. a. (2018) | $\sim$ 122K | 35 | 1024 | 1 | Image Segmentation |
| DeiT-Tiny-224 Touvron u. a. (2021) | Fractal-3K Kataoka u. a. (2022) | 3M | 80 | 32 | 16 | Image Classification |
| | (*) CIFAR-10 Krizhevsky und Hinton (2009) | 50K | 1000 | 8 | 96 | |
| | (*) CIFAR-100 Krizhevsky und Hinton (2009) | 50K | 1000 | 8 | 96 | |