# OpenReview forum: "KAKURENBO: Adaptively Hiding Samples in Deep Neural Network Training"
_NeurIPS.cc/2023/Conference — NeurIPS 2023 poster_

### Official Review · Reviewer_Zijr · 2023-07-02

**Soundness:** 2 fair
**Presentation:** 3 good
**Contribution:** 3 good
**Rating:** 6
**Confidence:** 4

**Summary:**

The paper introduces an approach to reduce training time of deep neural
networks on large datasets by excluding samples on an epoch basis with a
minimal reduction of the accuracy. In each epoch, a scheduled fraction of the
samples with the lowest loss is excluded, unless its prediction is incorrect
or its softmax confidence is below a fixed threshold. In addition, the learning
rate is scaled to account for the fraction of excluded samples.
The wall clock time as well as the accuracy of the introduced approach is
compared to four recent related approaches, as well as the unmodified training.

**Strengths:**

- The approach is novel, simple but seemingly effective.

- The approach is introduced very well.

- All figures are very high quality, especially the overview in Fig. 1.

- The paper provides a very good overview and contextualization over recent related methods (Table 1).

**Weaknesses:**

- The confidence value used for moving back samples is based on the model
  softmax, which is may not be very well calibrated.

- The experiments seem to be single trial. While training time makes this
  problematic, it is important to demonstrate that the results are significant.

**Questions:**

- The only major issue I see is that the experimental results seem to be based
  only on a single setup per trial. While the reason for this was most likely
  the runtime of the training, it is very important to demonstrate the
  significance of the results. If this cannot be adressed, I fear that this
  could be a reason to reject this otherwise very good work.

- The calibration of the softmax confidence should be discussed.


### Minor
- the textual references are odd (u.a. instead of et al., und instead of and)

**Limitations:**

The authors do not report any limitations.

---

> ### Author Rebuttal · Authors · 2023-08-10
>
> **5.1. Single setup per trial issue:**
> We confirm that the results of our implementation are stable by running experiments 3 times for CIFAR-100 and ImangetNet-1K. The new result is reported in Table 1 (extra-result in the rebuttal). As for our original results, due to resource limitation, we could not run experiments multiple times for each data point reported in the paper, especially for models and datasets that are big in this work.
>
> **5.2. The calibration of the softmax confidence:**
> As mentioned in the main manuscript, only samples that have low loss and sustain correct prediction with high confidence in the current epoch are hidden in the following epoch.  A sample is correctly predicted with high confidence at an epoch $e$ if it is predicted correctly (\textbf{PA}) and the prediction confidence (\textbf{PC}) is no less than a certain threshold \textbf{$\tau$}.
> As shown in Figure 1 (extra result in the rebuttal), the number of samples moved back becomes smaller when as the number of epochs increases because the prediction confidence becomes higher during the training, thus the actual hidden samples become similar to the max. hidden samples in the latter epoch. Results in Table 2 (extra result in the rebuttal) show that when we increase the threshold $\tau$ we obtain better accuracy (i.e., fewer hidden samples) at the cost of smaller performance improvements. However, the gaps remain small.
>
> **5.3. Textual references:** Thank you for the comments. We are happy to improve the text based on the suggestions.

---

> > ### Comment · Reviewer_Zijr · 2023-08-21
> > **Thank you for the rebuttal**
> >
> > Thank you for addressing all of my concerns.
> > The new experiments go into a good direction. I acknowledge that multiple trials for such large models are expensive, but note that if possible, the work should feature at least 20 trials, as statistical significance is critical to validate the results.
> > Nevertheless, given the new results and replies, I will increase my score to 6.

---

> ### Comment · Area_Chair_5DJW · 2023-08-20
>
> Dear Reviewer,
>
> Please respond to the author rebuttal as the discussion period is ending in less than 24 hours. The authors have put efforts into preparing the rebuttal and it would be best to respond to the rebuttal.
>
> Thanks,
> AC

---

### Official Review · Reviewer_gHmy · 2023-07-06

**Soundness:** 3 good
**Presentation:** 4 excellent
**Contribution:** 3 good
**Rating:** 6
**Confidence:** 4

**Summary:**

The paper proposes to accelerate DNN training by dynamically selecting a subset of the training set. The method records the per-sample loss during training, and hides (a heuristically set proportion of) samples with the least loss and correct prediction, then trains the model with an adjusted learning rate. The authors also provide a theoretical justification for the convergence. Experiments show that the proposed method reduces computation with negligible impact on the accuracy.

**Strengths:**

1.	The overall design is intuitive and technically sound. It is interesting to see this elegant framework using the simple hint of loss value.
2.	The authors provide a thorough summary of previous work, so it is clear for a wide community of readers to know this field.
3.	The manuscript is well organized. I like the logical overview before Sec 3.1.

**Weaknesses:**

1.	The hyperparameters (\tao, F_e) are heuristically set without analysis and discussion. It is unclear how many additional trials are required to obtain the claimed 20% acceleration. So I cannot make sure how it generalizes to untested datasets.
2.	We cannot see clearly (from the results) the superiority of the proposed method. It would be helpful to plot the accuracy-time trade-off of different baselines.
3.	There is a missing reference (Towards Sustainable Learning: Coresets for Data-efficient Deep Learning, ICML 2023) that targets the same problem with dynamic hiding, which claims a very good acceleration performance. A comparison would be helpful.

**Questions:**

1.	How does the hidden sample vary between epochs? It would be interesting to study how the contribution (to training) of samples goes ups and downs during learning.
2.	Do you think that per-sample gradients are a better metric for sorting? The gradient is also freely obtainable in training using back-propagation as the loss value.
3.	Can the method be combined with acceleration beyond data selection for a better performance? For example, Low dimensional trajectory hypothesis is true: DNNs can be trained in tiny subspaces, TPAMI 2022.

**Limitations:**

The authors do not discuss potential limitations in the manuscript.

---

> ### Author Rebuttal · Authors · 2023-08-10
>
> **4.1. Discussion on the hyperparameters ($\tau$, $F_e$):** Higher prediction confidence threshold $\tau$ leads to a higher number of samples moved back, i.e., fewer hidden samples at the beginning of the training process. At the end of the training process, when the model is well-trained, more samples are predicted correctly with high confidence. Thus the impact of the prediction confidence threshold on the number of moved-back samples becomes less (as shown in Figure 1 of the extra result in the rebuttal).  We also provide ablation study on the impact of prediction confidence threshold $\tau$ in Table 2 (extra result in the rebuttal). The result shows that when we increase the threshold $\tau$, wo obtain better accuracy (fewer hidden samples), but at the cost of smaller performance gain. We suggest to set $\tau = 0.7$ in all the experiments as a good trade-off between training time and accuracy.
>
> We consider the maximum fraction per epoch ($F_e$) as a hyperparameter and we suggest using step scheduling, i.e., to reduce the maximum hiding fraction gradually with a factor of $\alpha$ as the number of epochs increases. Specifically, we match the epoch at which $\alpha$ is reduced to the reduction of the learning rate in the literature. For example, we set $\alpha$ as [1, 0.8, 0.6, 0.4] at epoch numbers [0,30,60,80] for ImageNet-1K and [0, 60, 120, 180] for CIFAR-100, respectively.
>
>
>
> **4.2. Missing reference (Towards Sustainable Learning: Coresets for Data-efficient Deep Learning, ICML 2023)** The paper appear after our submission. We are happy to add this paper in the related work. Basically the proposed method in this paper has a high overhead of visiting each sample to find the coreset. In addition, it is only demonstrated on small datasets such as CIFAR and TinyImageNet, at which the overheads for a large number of samples would not appear. KAKURENBO on the other had is demonsrated on ImageNet-1K, DeepCAM, and ImageNet-1K-Fractal.
>
> **4.3. How does the hidden sample vary between epochs?**
> It is difficult to present how the contribution of each sample to the training process changes during training due to the large number of samples (in large datasets). Instead, in this study, we showed the contribution of each class to the training, which we partially discussed in Appendix C.1 (Figure 2) for ImageNet-1K datasets with ResNet-50. We also added new experimental results in Figure 2 (extra result in the rebuttal). The results show that the impact of each class remains different during the training. Easy classes such as class $1$, and $2$ are hidden during the training more than the class $31$ and $47$. The result in Figure 1 (extra result in the rebuttal) also shows that in two continuous epochs, only (around) 30\% of samples are hidden again. This result infers that the importance (or contribution level) of samples is changing epoch by epoch.
>
> **4.4. Do you think that per-sample gradients are a better metric for sorting?**
> Yes. However, the computation overhead for calculating the gradients of the hidden samples (performing both forward and backward pass on those samples) is not trivial. To reduce the overhead, we use the loss of samples that requires performing only the forward pass.
>
> **4.5. Can the method be combined with acceleration beyond data selection for a better performance?**
> To the best of our knowledge, we believe that KAKURENBO can be combined with the other loss-based optimizer and machine-learning methods. It also can be combined with other communication/computation optimization that have been proposed for speeding up the training process. More specifically, the referred paper by the reviewer proposed a quasi-Newton-based algorithm which is also based on the loss of samples for optimizing the model parameters. We believe that KAKURENBO can be combined with this method.
>
> **4.6. We cannot see clearly (from the results) the superiority of the proposed method. It would be helpful to plot the accuracy-time trade-off of different baselines.**
> We do plot the accuracy-time trade-off over different baselines (ISWR, FORGET, SB) in Figure 2 in the original submission.

---

> > ### Comment · Reviewer_gHmy · 2023-08-19
> > **Thanks for the rebuttal**
> >
> > Thanks for providing a good rebuttal. The authors have mostly addressed my concerns. Additional study on the contribution of each class to the training is interesting. I increased my rating. But I still think more numerical analysis on the hyperparameters (in addition to the rebuttal pdf) would be necessary to improve the manuscript.

---

> > > ### Author Response · Authors · 2023-08-19
> > >
> > > Thank you for the feed back. A full numerical analysis of the hyper parameters is certainly something we wanted to add to the rebuttal PDF, yet there is a single page restriction.

---

### Official Review · Reviewer_apYw · 2023-07-07

**Soundness:** 3 good
**Presentation:** 3 good
**Contribution:** 2 fair
**Rating:** 5
**Confidence:** 4

**Summary:**

This paper proposes a framework called KAKURENBO to accelerate DNN training by adaptively hiding samples. In the proposed framework, sample of low loss are hidden (i.e, not used) in the next training epoch, so that the total training steps are reduced. To compensate for the reduction in optimization steps, the authors propose to adjust the learning rate based on the ratio of hidden samples. Models trained with KAKURENBO achieves faster convergence with a little performance drop.

**Strengths:**

1. The paper is well organized and the logic is clear.
2. The proposed method achieves better convergence-accuracy tradeoff than some baseline methods.

**Weaknesses:**

1. The hidden factor F needs to be manually adjusted based on different datasets and training schedules.
2. The baseline of ImageNet-1K is a bit too weak for both ResNet and EfficientNet. EfficientNet-B3 achieves 81.6 Top-1 Acc in the original paper [1].
3. Lack of comparison with randomly hidding samples.
4. Lack of ablation study of not adjusting learning rate.
5. Lack of comparison with more advanced methods. For example, [2] also achieves fast convergence (saving ~30% training time), but without performance drop. The baseline of ImageNet-1K is also stronger in [2] (ResNet-50 76.4% in [2] vs 74.9 in this paper).

- [1] EfficientNet: Rethinking Model Scaling for Convolutional Neural Networks
- [2] InfoBatch: Lossless Training Speed Up by Unbiased Dynamic Data Pruning

**Questions:**

In Line 123, it is stated that "hidden samples that maintain a correct prediction with high confidence are moved back to the epoch training set.". Is it against the statement in Line 165 that "Only samples that have low loss and sustain correct prediction
with high confidence in the current epoch are hidden in the following epoch"?

**Limitations:**

Technical and experimental limitations as stated in weaknesses. No obvious societal negative impact.

---

> ### Author Rebuttal · Authors · 2023-08-10
>
> **3.1. The hidden factor F needs to be manually adjusted based on different datasets and training schedules**. We consider F as a hyperparameter, similarly to other methods in this domain. In our method, F is the maximum fraction, which can be adapted and changed during the training process.
>
> **3.2. The baseline of ImageNet-1K is a bit too weak for both ResNet and EfficientNet**. The baseline depends on hyper-parameters and the training technique used. We report the effectiveness of KAKURENBO with different training settings and accuracies in APPENDIX C.3 (Table 3 and Figure 4).
>
> **3.3. Lack of comparison with randomly hiding samples**
> We provide extra results in Table 1 to evaluate the training accuracy of random hiding with the CIFAR-100 dataset and WRN-28-10 model. As seen, accuracy is only 76.82\% which is lower than that of both KAKURENBO and Baseline. In fact, randomly hiding samples has been investigated before in the GradMatch paper and it's been reported that accuracy is low. This drove us originally not to evaluate this method.
>
> **3.4. Lack of ablation study of not adjusting learning rate.** Ablation study for not the adjusting learning rate is shown in APPENDIX C.4 (Table 5). The results show that if adjusting learning rate is not applied, the accuracy becomes lower.
>
> **3.5. Lack of comparison with more advanced methods**
> The paper in [1] appears after our submission. We are happy to add those papers to the related work.
> It is important to note that for method [1], the reported speedup is for a specific training regime (that uses a particular optimizer: LARS). The specific training regime described in the paper leads to a slow baseline (see Table 2 in [1], which shows the baseline that trains ImageNet-1K/ResNet-50 on 8 A100 GPUs for 90 epochs to be 3-4x slower than the typical number of hours to train ImageNet-1K/ResNet-50 on 8 A100 GPUs, as reported by many sources, including Nvidia NGC catalog [2]). That means while InfoBatch reports 26\% in speedup, the baseline setting for which InfoBatch is demonstrated to be effective is slow.
>
> [1] InfoBatch: Lossless Training Speed Up by Unbiased Dynamic Data Pruning)
> [2] \url{https://catalog.ngc.nvidia.com/orgs/nvidia/teams/dle/resources/resnet_pyt/performance}

---

> > ### Comment · Reviewer_apYw · 2023-08-19
> >
> > Thanks for the authors' response. Some of my concerns were addressed. But some problems still exist. As shown in Table.3 in the appendix, the performance drops non-trivially (76.56 -> 76.11) for the strong baseline, while the training time saved is limited (64060s ->  61723s). Experiments based on a low baseline are not so convincing. Overall, my judgment of this paper does not change and I would keep my rating.

---

> > > ### Author Response · Authors · 2023-08-19
> > >
> > > We would like to point out that per the results in Table 3 of the appendix, more time is saved when we cut up to 30% (dropping from 64060s to 59063s) while having the same accuracy when cutting up to 20%, i.e. we save ~10% of the time while dropping acc by <0.5%

---

> > > > ### Comment · Reviewer_apYw · 2023-08-21
> > > >
> > > > Thank you for your clarification. After re-consideration, I would like to raise the score to 5.

---

### Official Review · Reviewer_QXg1 · 2023-07-14

**Soundness:** 4 excellent
**Presentation:** 3 good
**Contribution:** 4 excellent
**Rating:** 7
**Confidence:** 3

**Summary:**

This paper introduces a method called KAKURENBO to increase the efficiency of training deep neural networks by reducing sample complexity while maintaining performance accuracy. The method encourages adaptive sample selection for each training epoch based on empirical criteria such as prediction accuracy (PA), prediction confidence (PC), and loss. It then adjusts the learning rate and maximum hidden fraction for each iteration to address convergence issues.

Additionally, the method reuses the empirical criteria (PC and loss) computed by the forward pass from the previous iteration as the basis for running the same selection mechanism in the next iteration.

The empirical evaluation of the proposed method against existing strategies on a range of datasets shows that KAKURENBO is effective in reducing total training time by up to 22%, with only a 0.4% impact on accuracy compared to the baseline.


**Strengths:**

$\textbf{Strong empirical result}$:
The results demonstrating the efficacy of KAKURENBO are compelling, as it outperformed other training strategies of the same type in terms of reducing training time while maintaining accuracy. The implementation details, including the source code, are thoroughly reported and appear to be reproducible. The ablation study is also conducted extensively. I believe that the proposed method would make a significant contribution to this subfield of research.


**Weaknesses:**

$\textbf{More complexity using a single GPU}$:
I notice that KAKURENBO takes more training time than the baseline, also with worse accuracy, using a single GPU on CIFAR-100 with ResNet-18 (Table 3). Could the authors provide the reasoning in more detail?

Minor redactional issues:
- L243: missing "." before "As the result ..."
- L287: accidental white space
- L276 - 284: please revisit the grammar


**Questions:**

No more except the one that I was asking in the “weaknesses” part.

**Limitations:**

Failure to generate more efficient training using 1 GPU should be properly addressed.

---

> ### Author Rebuttal · Authors · 2023-08-10
>
> **More complexity using a single GPU:**
> It is worth noting that KAKURENBO has computation overheads for updating the loss and prediction (Step D in Figure 1 in our main manuscript) and sorting the samples based on the loss (Step A in Figure 1 in our main manuscript). The performance gain from hiding samples will be limited if F is small, or potentially less than the overhead to compute the importance score of samples. In experiments using multiple GPUs, those operations are performed in parallel reducing the running time overhead. When using a single GPU on CIFAR-100 with ResNet-18 (Table 3 in our manuscript), the computational overhead is bigger than the speedup gained from hiding samples. Thus, KAKURENBO takes more training time in this case. In short, KAKURENBO is optimized for large-scale training and provides more benefits when running on multiple GPUs.

---

> > ### Comment · Reviewer_QXg1 · 2023-08-19
> >
> > Thank you for providing more context regarding the complexity of the training algorithm. Hoping that the corresponding explanation will be added into the manuscript as well.

---

### Official Review · Reviewer_wMxv · 2023-07-24

**Soundness:** 2 fair
**Presentation:** 2 fair
**Contribution:** 2 fair
**Rating:** 4
**Confidence:** 5

**Summary:**


Summary: This paper proposes a method of dynamically selecting training data during the training process to improve the efficiency of the model training. Specifically, the authors optimize the training process by dynamically adjusting sample weights, learning rates, and the upper limit of sample numbers. While the method is considered reasonable, it lacks surprising and novel features. The absence of experimental validation on the claimed jft-3b and lain-5b datasets somewhat diminishes the urgency for readers to accept and publish its findings.


**Strengths:**


(Positive) Table 1 demonstrates the differences between this paper and related literature effectively, which is commendable. Through comparisons and analysis, Table 1 vividly illustrates the innovation and differences of this paper's method from other approaches, helping readers gain a clearer understanding of the research's value and contributions.

(Positive) The dynamic adjustment of learning rate and sample number limit is interesting and reasonable. By dynamically adjusting these hyperparameters, the model can better adapt to changes in data and training progress, further optimizing the model's performance.

(Positive) The method in Section 3.3 is reasonable and also inspiring. This is a positive acknowledgment of the method in this section, indicating its value and providing readers with new ideas and insights.


(Positive and Negative) I am pleased to see the authors conducting experiments on DeepCAM. I hope to see the results of experiments on jft-3b and lain-5b datasets. This is an expectation for the authors' work as it will present a more comprehensive demonstration of the method's applicability and performance in different datasets and application scenarios.


**Weaknesses:**


(Negative) Please discuss the difference between this paper and data distillation. Data distillation typically focuses on using an auxiliary model to soften and distill training data to generate more representative data for optimizing the model's generalization ability. In contrast, the method proposed in this paper dynamically allocates weights to samples based on their losses and confidences during the training process, adjusting the influence of training data to enhance the model's training efficiency and performance.


(Negative) The authors mentioned the importance of their method on jft-3b and lain-5b datasets. I agree with this motivation. Therefore, I believe the authors should validate their method on these two datasets. Mentioning the importance of the method on datasets like jft-3b and lain-5b indeed highlights their motivation and research drive. Conducting thorough experiments on these datasets will strengthen the persuasiveness of the results, demonstrating the performance of the method in different datasets and application scenarios. Especially for jft-3b and lain-5b datasets, validating and verifying the method's effectiveness will further enhance the study's credibility and applicability.


(Negative) The authors' method only achieves a 22% improvement compared to traditional methods, which, in my view, is not significant. If the authors could increase the training time by 10 or 100 times, I think it would be very meaningful. Although a 22% improvement may not be considered remarkable, it is essential to consider that optimizing training time is not the sole objective. Achieving substantial speedup in larger training settings would still be meaningful and worthy of further exploration and research.

(Negative) Dynamically allocating weights based on loss and confidence for each sample is intuitive. Similar approaches exist in existing literature. Therefore, I do not find it particularly amazing. While the method of dynamically allocating weights based on loss and confidence may be intuitive and has similarities with existing literature, it might not be considered astonishing or groundbreaking in the field.


(Negative) Comparing Formula 2 and Formula 1, it is best for the authors to differentiate the contents of the summation subscripts to better represent their distinction between the two sampling methods. This is a specific suggestion for the authors' writing, ensuring accuracy and clarity in expression, making it easier for readers to understand the differences between the two sampling methods.


(Negative) As Sections 3.1 and 3.2 are so intuitive, I am not sure how to evaluate them. In summary, the methods in Sections 3.1 and 3.2 are reasonable but may not be awe-inspiring and admirable. This represents a neutral viewpoint on the paper's methods, acknowledging their rationality but suggesting that they may lack innovation and breakthroughs.

(Positive and Negative) I am pleased to see the authors conducting experiments on DeepCAM. I hope to see the results of experiments on jft-3b and lain-5b datasets. This is an expectation for the authors' work as it will present a more comprehensive demonstration of the method's applicability and performance in different datasets and application scenarios.



**Questions:**


See *Weaknesses

**Limitations:**


No. See *Weaknesses

---

> ### Author Rebuttal · Authors · 2023-08-10
>
> **1.1. Difference between this paper and data distillation**
> Data distillation reportedly introduces high overhead and is thus used mainly as a means to improve the model's generalization ability, and not reducing the amount of work. In comparison, the work in this paper aims to reduce the work by dynamically hiding samples, while sustaining the accuracy.
>
> **1.2. Validate KAKURENBO on jft-3b and lain-5b datasets**: JFT-3B is a closed dataset and LAION-5B is only links to files and not a dataset; there is no direct way to download it. We mention JFT-3B and LAION-5B in this paper as examples to motivate the work.
>
> **1.3 Comparing Formula 2 and Formula 1, it is best for the authors to differentiate the contents of the summation subscripts to better represent their distinction between the two sampling methods.}. We are happy to improve the text based on the suggestions. Specifically, we edit the term  $k\left(t\right)$ in the formation 2 in the manuscript to  $k_G\left(t\right)$ where $k_G\left(t\right)$ is sampled from samples that are not hidden.
>
> **1.4 No significant improvement in training time**:
> It is unattainable to speed-up the training process by 10-100X just by hiding samples during training as it would require 90-99\% of samples to be hidden without reducing accuracy. However, our method can be combined with other computation/communication optimization methods that have been proposed for speeding up training [1],[2]). Our method is helpful in pre-training a model with a huge dataset (e.g., in transfer learning). Training in such cases may have very high computing resource costs, e.g., 100s-1000s of GPU hours. Even a 22\% of reduction in training time would be helpful in reducing the cost of training in this case.
>
> [1] Nguyen, Truong Thao, François Trahay, Jens Domke, Aleksandr Drozd, Emil Vatai, Jianwei Liao, Mohamed Wahib, and Balazs Gerofi. "Why globally re-shuffle? Revisiting data shuffling in large scale deep learning." In 2022 IEEE International Parallel and Distributed Processing Symposium (IPDPS), pp. 1085-1096. IEEE, 2022.
>
> [2]Tang, Zhenheng, Shaohuai Shi, Xiaowen Chu, Wei Wang, and Bo Li. "Communication-efficient distributed deep learning: A comprehensive survey." arXiv preprint arXiv:2003.06307 (2020)

---

> > ### Comment · Reviewer_wMxv · 2023-08-22
> > **response to authors**
> >
> > -- Do the authors intend to convey that the LAION dataset is not currently available for download?
> >
> > -- May I kindly inquire if the mention of JFT-3B and LAION-5B serves solely as motivation for the work? Could this be widely considered as an overstatement?  Shouldn't the introduction section be rewritten to make it more scientifically supported rather than containing unsupported overclaims?

---

> > > ### Author Response · Authors · 2023-08-22
> > >
> > > Our intention of mentioning JFT-3B and LAION-5B was a sincere effort to give examples to the reader on the growth in datasets, which is a motivation for KAKURENBO. We acknowledge this could be misinterpreted as a list of datasets we empirically examine in the paper. We appreciate the feedback on this point.
> > >
> > > We commit to to updating the introduction to be give a general statement about the rise in dataset sizes, and only explicitly mention the large datasets we examine in this paper: ImageNet-1K (1M images), Fractal-1K pre-training on ViT (3M images), and DeepCam (122K images of very large-sized images).
> > >
> > >
> > > >-- Do the authors intend to convey that the LAION dataset is not currently available for download?
> > >
> > > LAION does not provide the dataset in that form that is prepared and compressed in a single location for download. LAION only provides links to the images. Downloading the dataset requires individual downloading of every single image from its source host (a total of 240TB), which is proving to be challenging: can take months to download. We currently don't have to capacity to download, store, and inspect the 240TB dataset. That being said, there are several community efforts around efficient downloaders and caching, which we hope to be able to use.

---

> > > > ### Comment · Reviewer_wMxv · 2023-08-22
> > > > **response to authors**
> > > >
> > > > Thanks for the reply!

---

> ### Comment · Area_Chair_5DJW · 2023-08-20
>
> Dear Reviewer,
>
> Please respond to the author rebuttal as the discussion period is ending in less than 24 hours. The authors have put efforts into preparing the rebuttal and it would be best to respond to the rebuttal.
>
> Thanks,
> AC

---

### Author Rebuttal · Authors · 2023-08-10

We thank all reviewers for their constructive comments and questions.

**1. Single setup per trial issue (Reviewer Zijr)**
We confirm that the results of our implementation are stable by running experiments 3 times for CIFAR-100 and ImangetNet-1K. The new result is reported in Table 1 (extra-result in the rebuttal). As for our original results, due to resource limitations, we could not run experiments multiple times for each data point reported in the paper, especially for models and datasets that are big in this work.

**2. Discussion on the hyperparameters (Reviewer gHmy and Zijr)**
Higher prediction confidence threshold $\tau$ leads to a higher number of samples being moved back to the training set, i.e., fewer hidden samples at the beginning of the training process. At the end of the training process, when the model has is well-trained, more samples are predicted correctly with high confidence. Thus the impact of the prediction confidence threshold on the number of moved-back samples becomes less (as shown in Figure 1 of the extra result in the rebuttal).  We also provide ablation study on the impact of prediction confidence threshold $\tau$ in Table 2 (extra result in the rebuttal). The result shows that when we increase the threshold $\tau$, we obtain better accuracy (fewer hidden samples), but at the cost of smaller performance gain. We suggest to set $\tau = 0.7$ in all the experiments as a good trade-off between training time and accuracy.

We consider the maximum fraction per epoch ($F_e$) as a hyperparameter and we suggest using step scheduling, i.e., to reduce the maximum hiding fraction gradually with a factor of $\alpha$ as the number of epochs increases. Specifically, we match the epoch at which $\alpha$ is reduced to the reduction of the learning rate in the literature. For example, we set $\alpha$ as [1, 0.8, 0.6, 0.4] at epoch numbers [0,30,60,80] for ImageNet-1K and [0, 60, 120, 180] for CIFAR-100, respectively.

**3. Comparison with other methods (Reviwer apYm and Zijr)**
We provide extra results in Table 1 %(extra results in the rebuttal)
to evaluate the training accuracy of random hiding with the CIFAR-100 dataset and WRN-28-10 model. As seen, accuracy is only 76.82\% which is lower than that of both KAKURENBO and Baseline. In fact, randomly hiding samples has been investigated before in the GradMatch paper and it's been reported that accuracy is low. This drove us originally not to evaluate this method.

For the works in [1], and [2], both of the papers appear after our submission. We are happy to add those papers to the related work.
It is important to note that for method [1], the reported speedup is for a specific training regime (that uses a particular optimizer: LARS). The specific training regime described in the paper leads to a slow baseline (see Table 2 in [1], which shows the baseline that trains ImageNet-1K/ResNet-50 on 8 A100 GPUs for 90 epochs to be 3-4x slower than the typical number of hours to train ImageNet-1K/ResNet-50 on 8 A100 GPUs, as reported by many sources, including Nvidia NGC catalog [3]). That means while InfoBatch reports 26\% in speedup, the baseline setting for which InfoBatch is demonstrated to be effective is slow. The proposed method in [2] has a high overhead of visiting each sample to find the coreset.

[1] InfoBatch: Lossless Training Speed Up by Unbiased Dynamic Data Pruning)

[2] (Towards Sustainable Learning: Coresets for Data-efficient Deep Learning, ICML 2023)

[3] \url{https://catalog.ngc.nvidia.com/orgs/nvidia/teams/dle/resources/resnet_pyt/performance}

**4. Improvement in training time (Reviewer wMxv, QXg1, and gHmy)**
It is worth noting that KAKURENBO has computation overheads for updating the loss and prediction (Step D in Figure 1 in our main manuscript) and sorting the samples based on the loss (Step A in Figure 1 in our main manuscript). The performance gain from hiding samples will be limited if F is small, or potentially less than the overhead to compute the importance score of samples. In experiments using multiple GPUs, those operations are performed in parallel reducing the running time overhead. When using a single GPU on CIFAR-100 with ResNet-18 (Table 3 in our manuscript), the computational overhead is bigger than the speedup gained from hiding samples. Thus, KAKURENBO takes more training time in this case. In short, KAKURENBO is optimized for large-scale training and provides more benefits when running on multiple GPUs.

It is unattainable to speed-up the training process by 10-100X just by hiding samples during training as it would require 90-99\% of samples to be hidden without reducing accuracy. However, our method can be combined with other computation/communication optimization methods that have been proposed for speeding up training [4],[5]). Our method is helpful in pre-training a model with a huge dataset (e.g., in transfer learning). Training in such cases may have very high computing resource costs, e.g., 100s-1000s of GPU hours. Even a 22\% of reduction in training time would be helpful in reducing the cost of training in this case.

[4] Nguyen, Truong Thao, François Trahay, Jens Domke, Aleksandr Drozd, Emil Vatai, Jianwei Liao, Mohamed Wahib, and Balazs Gerofi. "Why globally re-shuffle? Revisiting data shuffling in large scale deep learning." In 2022 IEEE International Parallel and Distributed Processing Symposium (IPDPS), pp. 1085-1096. IEEE, 2022.

[5]Tang, Zhenheng, Shaohuai Shi, Xiaowen Chu, Wei Wang, and Bo Li. "Communication-efficient distributed deep learning: A comprehensive survey." arXiv preprint arXiv:2003.06307 (2020).

**5. The baseline of ImageNet-1K is a bit too weak (Reviewer apYw)**
The baseline depends on hyper-parameters and the training technique used. We report the effectiveness of KAKURENBO with different training settings and accuracies in APPENDIX C.3 (Table 3 and Figure 4).

---

### Comment · Area_Chair_5DJW · 2023-08-13
**Please respond to author rebuttals**

Dear Reviewers,

Please respond to authors after carefully reading the author rebuttals and other reviews. If your assessment of the paper changes, please update your score with a short justification for the new rating.

The paper received diverging initial reviews. Please consider discussing with the authors or other reviewers to see whether we can reach a consensus.

Thank you,
AC

---

### Decision · Program_Chairs · 2023-09-21

**Decision:**

Accept (poster)

**Comment:**

The paper proposes a method to adaptively remove samples based on their prediction accuracy, confidence and loss. The ratings are 1 accept, 2 weak accepts, 1 borderline accept and 1 borderline reject. The leaning negative reviewer has the remaining concern about the mentioning of LAION and JFT datasets as motivating examples in their final comment, but the AC views this as fine. Reviewers agree the results are favorable compared with existing methods, and most reviewers agree that the method is intuitive and clear. The AC also agrees that the addressed problem about data selection is important. These outweigh the potential issues brought by reviewers, but the authors are also encouraged to take these suggestions to improve the paper. The AC is pleased to recommend the acceptance of this paper.